# Multispecies totally asymmetric simple exclusion process with long-range swap

**Eunghyun Lee**[*]
Department of Mathematics,
School of Sciences and Humanities,
Nazarbayev University,
Kazakhstan

### Abstract

We introduce the multispecies totally asymmetric simple exclusion process (mTASEP) with long-range swap, a new interacting particle system combining the backward-push rule with the forward-jump rule. Although governed by local dynamics, the model induces effective long-range particle exchanges. We establish its integrability by proving two-particle reducibility and showing that the associated scattering matrix satisfies the Yang–Baxter equation. In addition, we derive explicit contour integral formulas for transition probabilities. These results position the long-range swap model as a novel exactly solvable multispecies process, characterized by distinctive algebraic features and opening new directions for further study in integrable probability and statistical mechanics.

## 1  Introduction

Interacting particle systems have long served as fundamental models in probability theory and statistical mechanics, providing both exactly solvable examples and rich universality phenomena. Among them, the totally asymmetric simple exclusion process (TASEP) occupies a central role, being one of the simplest nontrivial stochastic models of transport with broad connections to combinatorics, integrable systems, and the Kardar–Parisi–Zhang (KPZ) universality class [5, 7, 9, 10, 17, 21, 23]. Over the past decades, various generalizations have been proposed to capture more complex interaction rules, including multispecies extensions [11, 25], inhomogeneous jumping rates [4, 26], particle/species-dependent rates [14, 20] and families with one- or two-parameter deformations [2, 3, 19]. These variants not only broaden the family of integrable stochastic models but also reveal novel algebraic structures and probabilistic behaviors. In this work, we introduce and analyze a new integrable multispecies model with long-range swap dynamics, and place it within the broader landscape of TASEP-type and drop–push-type dynamics [22].

---

[*]eunghyun.lee@nu.edu.kz

We adopt the convention used in [13–16] to describe the state of the multispecies interacting particle system on $\mathbb{Z}$ with the exclusion rule. A state of the $n$-particle system is represented by a pair $(X, \pi) = (x_1, \ldots, x_n, \pi_1 \cdots \pi_n)$, where $X = (x_1, \ldots, x_n) \in \mathbb{Z}^n$ denotes the positions of the particles, ordered such that $x_1 < \cdots < x_n$. Thus, $x_i$ is the position of the $i$th leftmost particle. The sequence $\pi = \pi_1 \pi_2 \cdots \pi_n$ is a word of length $n$ with entries in $\{1, \ldots, N\}$, indicating that $\pi_i$ denotes the species of the $i$th leftmost particle.

In the classical multispecies totally asymmetric simple exclusion process (mTASEP), particles evolve in continuous time. Each particle at $x$ waits an exponential time with rate 1 before attempting a rightward move to $x + 1$. If the target site is vacant, the particle occupies it. If the site is occupied by another particle, the interaction depends on their species labels. When a particle of species $i$ attempts to move into a site occupied by species $j$, the move succeeds with an exchange of positions if $i > j$, while it is suppressed if $i \leq j$.

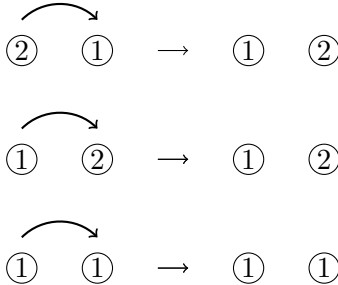

Figure 1: Illustration of the mTASEP dynamics. In the second and third figures, the configuration remains unchanged.

The notion of *hidden states*, introduced in [16], provides a convenient reformulation of the mTASEP dynamics. A hidden state is a transient configuration in which a site momentarily accommodates two particles side by side before immediately transitioning to another state. If site $x$ holds particles of species $i$ (left) and $j$ (right), we denote this hidden state by $(x, x, ij)$. Since hidden states vanish instantaneously, they are not in the Markov state space. Using this perspective, the dynamics of the mTASEP can be described as follows: when a particle of species $i$ at site $x - 1$ attempts to move onto a particle of species $j$ at site $x$, the particle $i$ jumps to site $x$ and occupies the left position there, creating the hidden state $(x, x, ij)$. A position adjustment then occurs: the particle of species $\min(i, j)$ remains on the left at $x$, and then it immediately jumps back to site $x - 1$. This hidden state formalism enables a fully local description of the update rules and greatly facilitates algebraic treatment of the model.

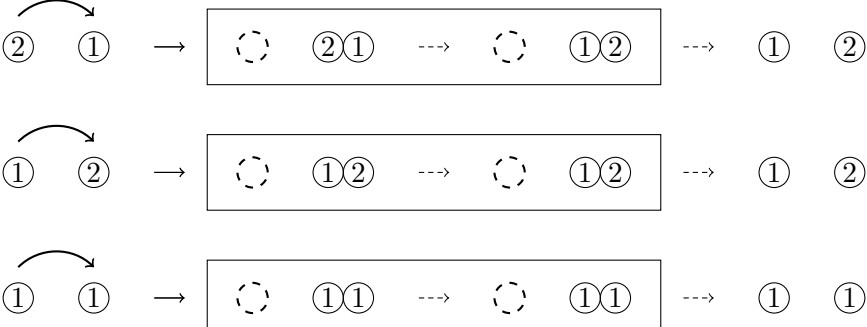

Figure 2: Hidden-state representation of the mTASEP dynamics shown in Figure 1. Dashed arrows denote an instantaneous transition.

From the particle's perspective, the mTASEP rules may be summarized as follows: a stronger particle entering a site occupied by a weaker one displaces the weaker one backward, while a weaker particle entering a site occupied by a stronger particle jumps back to the original site. This pair of interactions, "backward push" and "backward jump", characterizes the mTASEP.

A different variant of the multispecies model, the long-range push model, introduced in [15], is defined by the rule that a particle of species $i$ jumps to the nearest site to its right that is occupied by a weaker particle $j < i$ (with $j = 0$ denoting a vacancy), whereupon $i$ pushes $j$ forward. (Its single-species version is known as the drop–push model [22].) The displaced particle $j$ then follows the same rule, which can trigger a cascade of displacements. Interpreted in terms of hidden states, this mechanism may be reformulated as a sequence of local updates: if a particle of species $i$ at site $x - 1$ attempts to move onto a particle of species $j$ at site $x$, then $i$ jumps to site $x$ and occupies the left position, forming the hidden state $(x, x, ij)$. A position adjustment then occurs: the particle of species $\min(i, j)$ is placed on the right at $x$ and immediately jumps forward to $x + 1$. If $x + 1$ is already occupied, the same rule applies recursively, producing a chain of displacements. In this model, therefore, a stronger particle entering a site occupied by a weaker particle pushes the weaker one forward, while a weaker particle entering a site occupied by a stronger particle jumps forward over the stronger particle. This pairing of "forward push" and "forward jump" yields another exactly solvable interacting particle system.

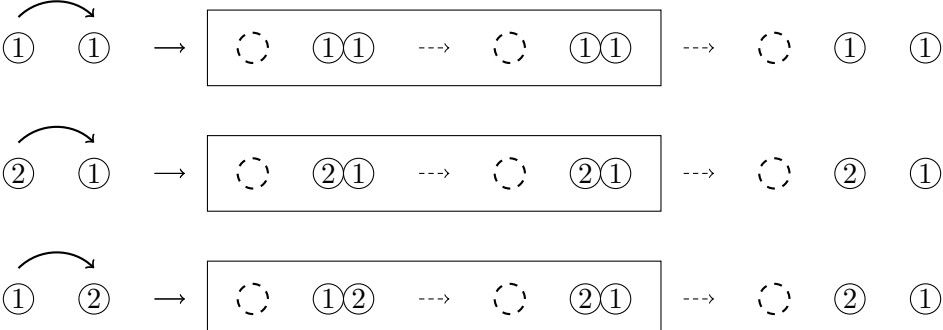

Figure 3: Illustration of the dynamics of the long-range push model

These two established dynamics naturally raise the question of what happens when the interaction rules are combined in a hybrid fashion. For example, one may consider a system where stronger particles push weaker ones backward while weaker particles jump forward over stronger ones, or, conversely, a system where stronger particles push weaker ones forward while weaker ones jump backward when they encounter stronger ones. As we demonstrate, the first hybrid leads precisely to the long-range swap model studied in this paper, which turns out to be integrable and structurally rich, whereas the second fails to yield a consistent multi-particle dynamics and is therefore non-integrable.

Integrability of an interacting particle system in one dimension requires two key ingredients. First, the system must satisfy two-particle reducibility: the master equation for any configuration can be reduced to the evolution equation not involving the particle interaction, supplemented by boundary conditions involving only two-particle interactions. For our model, verifying this reducibility is significantly more intricate than in the mTASEP or in the long-range push model [13, 15]. Second, the associated scattering matrix must obey the Yang–Baxter relation, ensuring the consistency of multi-particle scattering and thus the solvability of the model via the Bethe ansatz.

The remainder of the paper is organized as follows. In Section 2 we define the model precisely and present its nearest-neighbor formulation, which realizes long-range swaps through local moves. Section 3 establishes the model's integrability, proving two-particle reducibility and Yang–Baxter consistency, and derives contour-integral formulas for the transition probabilities. Section 4 provides a discussion of structural features, followed by a summary and outlook in Section 5.

# 2   Definition of the model

## 2.1   Notations

We follow the conventions of [13, 15, 16]. For an initial configuration $(Y, \nu)$ and terminal configuration $(X, \pi)$ after time $t$, let

$$P_{(Y,\nu)}(X, \pi; t)$$

denote the transition probability. We write $\mathbf{P}(t)$ for the infinite matrix whose columns are indexed by initial states $(Y, \nu)$ and rows by terminal states $(X, \pi)$ at time $t$. The $((X, \pi), (Y, \nu))$-entry of $\mathbf{P}(t)$ is exactly $P_{(Y,\nu)}(X, \pi; t)$.

For fixed $X$ and $Y$, let $\mathbf{P}_Y(X; t)$ be the $N^n \times N^n$ submatrix of $\mathbf{P}(t)$ obtained by restricting to rows indexed $(X, \pi)$ and columns indexed $(Y, \nu)$, with $\pi, \nu$ ranging over all words of length $n$ with letters in $\{1, \ldots, N\}$. Unless otherwise specified, the rows and columns of all $N^n \times N^n$ matrices in this paper are ordered lexicographically, from $11 \cdots 1$ to $NN \cdots N$.

When the initial state $(Y, \nu)$ is clear from context or immaterial to the discussion, we abbreviate $\mathbf{P}_Y(X; t)$ as $\mathbf{P}(X; t)$, and similarly $P_{(Y,\nu)}(X, \pi; t)$ as $P_\nu(X, \pi; t)$ or simply $P(X, \pi; t)$. For such matrices, derivatives are taken entrywise

$$\left(\frac{d}{dt}\mathbf{P}(X; t)\right)_{\pi,\nu} := \frac{d}{dt}P_\nu(X, \pi; t).$$

## 2.2   Definition of the Dynamics

**Definition 2.1.** Consider a continuous-time interacting particle system on $\mathbb{Z}$ with species labels in $\{1, \ldots, N\}$. Each particle of species $i$ waits an exponential time with rate 1, and then exchanges positions with the nearest weaker particle $j < i$ to its right (with the convention that an empty site is treated as a particle of species 0). We call this process the **multispecies totally asymmetric simple exclusion process with long-range swap** (abbreviated as *mTASEP with long-range swap*, or simply the *long-range swap model*).

Although phrased in terms of long-range exchanges, this dynamics can be realized entirely through local nearest-neighbor updates. Indeed, combining the backward push rule (from the mTASEP) with the forward jump rule (from the long-range push model) allows long-range swaps to be decomposed into consecutive local interactions..

Here are the descriptions of the two local rules with illustrations through hidden states for the mTASEP with long-range swap:

- **Backward push by a stronger particle**: If a particle of species $i$ jumps into a site occupied by a weaker species $j$ with $j < i$, then the weaker particle is displaced back to the site

vacated by $i$, producing an exchange of positions.

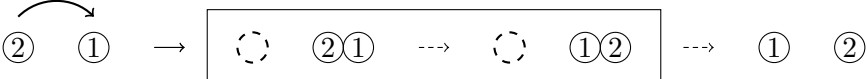

Figure 4: Particle of species 2 pushes particle of species 1 backward.

- **Forward jump by a weaker particle**: If a particle of species $i$ jumps into a site occupied by a stronger species $j$ with $j \geq i$, then $i$ immediately skips over $j$ to the next site in the same direction. (If a weaker particle is pushed leftward to a site occupied by a stronger particle, the same "jump-over" rule applies to the left as well).

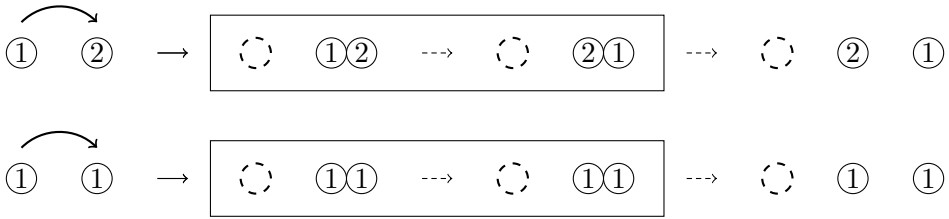

Figure 5: Particle of species $i$ jumps over particle of species $j \geq i$ to the next site in the direction of motion.

**Example 2.1.** Suppose that the particle of species 2 jumps in the state $(x, x+1, x+2, 231)$. By Definition 2.1, the particle of species 2 performs a long-range swap with the particle of species 1,

$$(x, x+1, x+2, 231) \rightarrow (x, x+1, x+2, 132).$$

This transition can be decomposed into a sequence of local nearest-neighbor updates as follows:

(1) The particle of species 2 first jumps forward over the particle of species 3 at $x+1$, reaching $x+2$.

Figure 6: Illustration of transition $(x, x+1, x+2, 231) \rightarrow (x+1, x+2, x+2, 321)$

(2) There it meets the particle of species 1, which is pushed backward to $x+1$.

Figure 7: Illustration of transition $(x+1, x+1, x+2, 321) \rightarrow (x+1, x+1, x+2, 312)$

(3) The displaced particle of species 1 then encounters the particle of species 3 at $x+1$ and jumps over it to $x$.

Figure 8: Illustration of transition $(x + 1, x + 2, x + 2, 312) \to (x, x + 1, x + 2, 132)$

This chain of local updates realizes a long-range swap.

**Remark 2.1.** In this formulation, two particles of the same species interact as if one were stronger: the arriving particle treats the other as a stronger species as in the drop-push model.

## 2.3  Two-Particle Interaction Matrices

In the case of the mTASEP, the interaction of two particles at a common site $x$ as illustrated in the boxes in Figure 2, can be encoded in a $4 \times 4$ matrix whose rows and columns are labelled $11, 12, 21, 22$:

$$
\begin{array}{c}
 \\
11 \\
12 \\
21 \\
22
\end{array}
\begin{array}{cccc}
11 & 12 & 21 & 22 \\
\left( \begin{array}{cccc}
1 & 0 & 0 & 0 \\
0 & 1 & 1 & 0 \\
0 & 0 & 0 & 0 \\
0 & 0 & 0 & 1
\end{array} \right),
\end{array}
\tag{1}
$$

where the $(ij, kl)$-entry represents the probability of the transition from the hidden state $(x, x, kl)$ to $(x, x, ij)$. This interaction rule at a common site, combined with the rule for the backward move of a weaker particle is expressed in the Bethe ansatz formulation by the boundary condition

$$
\mathbf{U}(x, x; t) = \begin{pmatrix}
1 & 0 & 0 & 0 \\
0 & 1 & 1 & 0 \\
0 & 0 & 0 & 0 \\
0 & 0 & 0 & 1
\end{pmatrix} \mathbf{U}(x, x + 1; t)
$$

as detailed in [12]. Here, $\mathbf{U}(x_1, x_2; t) = (u_{\pi,\nu})$ denotes a $4 \times 4$ matrix where $u_{\pi,\nu}$ is a function $U_\nu(x_1, x_2, \pi; t)$ defined for all $(x_1, x_2) \in \mathbb{Z}^2$ and $t \geq 0$.

For the mTASEP with long-range push, in which a weaker particle jumps forward from a common site, the boundary condition is given by

$$
\mathbf{U}(x, x; t) = \begin{pmatrix}
1 & 0 & 0 & 0 \\
0 & 0 & 0 & 0 \\
0 & 1 & 1 & 0 \\
0 & 0 & 0 & 1
\end{pmatrix} .\mathbf{U}(x - 1, x; t)
$$

(See [15] for the details.)

We now derive the boundary condition for the mTASEP with long-range swap, which mixes the mTASEP rule and the long-range push rules. The local transition rules between hidden states at a common state, shown in the boxes in Figure 4 and 5, can be encoded by the matrix

$$
\begin{array}{c}
\phantom{11} \begin{array}{cccc} 11 & 12 & 21 & 22 \end{array} \\
\begin{array}{c} 11 \\ 12 \\ 21 \\ 22 \end{array}
\left(
\begin{array}{cccc}
1 & 0 & 0 & 0 \\
0 & 0 & 1 & 0 \\
0 & 1 & 0 & 0 \\
0 & 0 & 0 & 1
\end{array}
\right).
\end{array}
\tag{2}
$$

However, this matrix splits into two parts in the boundary condition, as we now demonstrate. For example, the master equations of $P(x, x+1, 12; t)$ and $P(x, x+1, 21; t)$ are

$$
\frac{d}{dt} P(x, x+1, 12; t) = P(x-1, x+1, 12; t) + P(x, x+1, 21; t) - 2P(x, x+1, 12; t),
$$

and

$$
\frac{d}{dt} P(x, x+1, 21; t) = P(x-1, x+1, 21; t) + P(x-1, x, 12; t) - 2P(x, x+1, 21; t),
$$

respectively. Encoding the master equations of $P_\nu(x, x+1, \pi; t)$ for all $\pi, \nu \in \{11, 12, 21, 22\}$ in a $4 \times 4$ matrix form yields

$$
\frac{d}{dt} \mathbf{P}(x, x+1; t)
$$

$$
= \mathbf{P}(x-1, x+1; t) +
\left(
\begin{array}{cccc}
1 & 0 & 0 & 0 \\
0 & 0 & 0 & 0 \\
0 & 1 & 0 & 0 \\
0 & 0 & 0 & 1
\end{array}
\right) \mathbf{P}(x-1, x; t) +
\left(
\begin{array}{cccc}
0 & 0 & 0 & 0 \\
0 & 0 & 1 & 0 \\
0 & 0 & 0 & 0 \\
0 & 0 & 0 & 0
\end{array}
\right) \mathbf{P}(x, x+1; t)
\tag{3}
$$

$$
- 2\mathbf{P}(x, x+1; t).
$$

Similarly, for $x_1 < x_2 - 1$, we have

$$
\frac{d}{dt} \mathbf{P}(x_1, x_2; t) = \mathbf{P}(x_1 - 1, x_2; t) + \mathbf{P}(x_1, x_2 - 1; t) - 2\mathbf{P}(x_1, x_2; t).
\tag{4}
$$

Let $\mathbf{U}(x_1, x_2; t) = (u_{\pi, \nu})$ denotes a $4 \times 4$ matrix, where each entry $u_{\pi, \nu}$ is given by a function $U_\nu(x_1, x_2, \pi; t)$, defined for all $(x_1, x_2) \in \mathbb{Z}^2$ and $t \geq 0$. Suppose that $\mathbf{U}(x_1, x_2; t)$ satisfies

$$
\frac{d}{dt} \mathbf{U}(x_1, x_2; t) = \mathbf{U}(x_1 - 1, x_2; t) + \mathbf{U}(x_1, x_2 - 1; t) - 2\mathbf{U}(x_1, x_2; t)
\tag{5}
$$

for all $(x_1, x_2) \in \mathbb{Z}^2$ together with the *boundary condition*

$$
\mathbf{U}(x, x; t) =
\left(
\begin{array}{cccc}
1 & 0 & 0 & 0 \\
0 & 0 & 0 & 0 \\
0 & 1 & 0 & 0 \\
0 & 0 & 0 & 1
\end{array}
\right) \mathbf{U}(x-1, x; t) +
\left(
\begin{array}{cccc}
0 & 0 & 0 & 0 \\
0 & 0 & 1 & 0 \\
0 & 0 & 0 & 0 \\
0 & 0 & 0 & 0
\end{array}
\right) \mathbf{U}(x, x+1; t)
\tag{6}
$$

for all $x \in \mathbb{Z}$. Then, it is straightforward that $\mathbf{U}(x_1, x_2; t)$ satisfies the master equation (3) and (4) for $X = (x, x+1)$ and $X = (x_1, x_2)$ with $x_1 < x_2 - 1$, respectively.

**Definition 2.2.** Let $\mathbf{B} = (b_{\pi,\nu})$ and $\mathbf{B}' = (b'_{\pi,\nu})$ be the $N^2 \times N^2$ matrices whose rows and columns are indexed lexicographically by words $\pi = \pi_1 \pi_2$ and $\nu = \nu_1 \nu_2$, respectively, from $11$ to $NN$. Their entries are defined by

$$b_{\pi,\nu} = \begin{cases} 1 & \text{if } \pi = \nu \text{ and } \nu_1 = \nu_2, \text{ or } \pi_1 = \nu_2, \pi_2 = \nu_1 \text{ and } \nu_1 < \nu_2 \\ 0 & \text{otherwise;} \end{cases}$$

$$b'_{\pi,\nu} = \begin{cases} 1 & \text{if } \pi_1 = \nu_2, \pi_2 = \nu_1 \text{ and } \nu_1 > \nu_2 \\ 0 & \text{otherwise,} \end{cases}$$

respectively.

Thus, the first and second matrices in (6) correspond precisely to $\mathbf{B}$ and $\mathbf{B}'$ with $N = 2$, respectively. Here, $b_{\pi,\nu}$ describes the rate of a transition from state $(x-1, x, \nu_1\nu_2)$ to $(x, x+1, \pi_1\pi_2)$ caused by the jump of particle $\nu_1$, while $b'_{\pi,\nu}$ describes the rate of a transition from state $(x, x+1, \nu_1\nu_2)$ to $(x, x+1, \pi_1\pi_2)$ caused by the jump of particle $\nu_1$.

## 2.4 Left-Right Symmetry of the Boundary Condition

At the outset, we assumed that particles move to the right after their waiting times, and that when entering a site occupied by another particle of the same species, the incoming particle treats the resident as stronger. If instead we assume leftward motion and that an incoming particle treats a same-species resident as weaker, we obtain the same boundary condition as in (6)

In the right-moving system, the first matrix in (6) corresponds to the forward jump of a weaker particle over a stronger particle, while the second matrix corresponds to a stronger particle pushing a weaker particle backward. In the left-moving system, the roles are reversed: the second matrix corresponds to the forward jump of a weaker particle over a stronger particle, and the first matrix corresponds to a stronger particle pushing a weaker particle backward.

We also note that in the right-moving system, a particle may be forced to jump over a stronger particle to the left due to the backward push rule. This situation is naturally captured by the second matrix in (6).

# 3 Integrability

## 3.1 Two-particle interaction reducibility

### 3.1.1 Master equations for three-particle system

For solvability via the Bethe ansatz, the master equation of an $n$-particle system must reduce to a combination of the evolution equation for non-interacting particles together with boundary conditions that depend only on pairwise interactions. This property is often referred to as two-particle reducibility. In the case $n = N = 2$, we already observed in Section 2.3 that the master equation for two particles can be expressed as the evolution equation (5), which governs free particle motion (i.e., without interactions), together with the boundary condition (6).

To test whether the same principle extends to larger systems, we now examine the case $n = N = 3$. Compared to the mTASEP and the long-range push model [13,15,16], the mTASEP with long-range swap requires more intricate analysis. We analyze each configuration systematically, distinguishing situations where particles are well separated from those where they occupy adjacent sites.

(i) First, it is clear that for $X = (x_1, x_2, x_3)$ with $x_i < x_{i+1} - 1$, $(i = 1, 2)$,

$$\frac{d}{dt}\mathbf{P}(X;t) = \mathbf{P}(x_1 - 1, x_2, x_3; t) + \mathbf{P}(x_1, x_2 - 1, x_3; t) + \mathbf{P}(x_1, x_2, x_3 - 1; t) - 3\mathbf{P}(X;t), \quad (7)$$

where we recall that the matrices $\mathbf{P}$ are $27 \times 27$ matrices.

(ii) For $X = (x, x + 1, x_3)$ with $x + 1 < x_3 - 1$, the master equation of $\mathbf{P}(X;t)$ takes the form

$$\begin{aligned}
\frac{d}{dt}\mathbf{P}(X;t) &= \mathbf{P}(x - 1, x + 1, x_3; t) + \mathbf{P}(x, x + 1, x_3 - 1; t) \\
&\quad + \mathbf{M}\,\mathbf{P}(x - 1, x, x_3; t) + \mathbf{M}'\,\mathbf{P}(x, x + 1, x_3; t) - 3\mathbf{P}(X;t)
\end{aligned} \quad (8)$$

where $\mathbf{M} = (m_{\pi,\nu})$ and $\mathbf{M}' = (m'_{\pi,\nu})$ are $27 \times 27$ matrices.

Following the arguments in [15, 16], we claim that

$$\mathbf{M} = \mathbf{B} \otimes \mathbf{I}, \qquad \mathbf{M}' = \mathbf{B}' \otimes \mathbf{I},$$

where $\mathbf{I}$ is the $3 \times 3$ identity matrix, $\otimes$ denotes the tensor (Kronecker) product of matrices, and $\mathbf{B}, \mathbf{B}'$ are the $3^2 \times 3^2$ matrices defined in Definition 2.2.

Indeed, $m_{\pi,\nu}$ gives the rate of the transition

$$(x - 1, x, x_3, \nu_1\nu_2\nu_3) \longrightarrow (x, x + 1, x_3, \pi_1\pi_2\pi_3),$$

while $m'_{\pi,\nu}$ gives the rate of the transition

$$(x, x + 1, x_3, \nu_1\nu_2\nu_3) \longrightarrow (x, x + 1, x_3, \pi_1\pi_2\pi_3).$$

In the first case, the third particle neither moves nor interacts, considering the change in the position configuration, so $\pi_3 = \nu_3$ and the transition rate depends only on the first two particles, exactly as in the two-particle system. Hence, using the relation

$$(\mathbf{B} \otimes \mathbf{I})_{\pi_1 \pi_2 \pi_3, \nu_1 \nu_2 \nu_3} = \begin{cases} (\mathbf{B})_{\pi_1 \pi_2, \nu_1 \nu_2}, & \text{if } \pi_3 = \nu_3, \\ \\ 0, & \text{if } \pi_3 \neq \nu_3, \end{cases}$$

we conclude that $\mathbf{M} = \mathbf{B} \otimes \mathbf{I}$. Analogous arguments give $\mathbf{M}' = \mathbf{B}' \otimes \mathbf{I}$.

(iii) By the same reasoning as in the case $X = (x, x+1, x_3)$ above, the master equation for $\mathbf{P}(X; t)$ in the case $X = (x_1, x, x+1)$ with $x_1 < x - 1$ is

$$\frac{d}{dt}\mathbf{P}(X; t) = \mathbf{P}(x_1 - 1, x, x+1; t) + \mathbf{P}(x_1, x-1, x+1; t) \tag{9}$$
$$+ (\mathbf{I} \otimes \mathbf{B})\,\mathbf{P}(x_1, x-1, x; t) + (\mathbf{I} \otimes \mathbf{B}')\,\mathbf{P}(x_1, x, x+1; t) - 3\mathbf{P}(X; t).$$

(iv) Finally, consider the case $X = (x, x+1, x+2)$. In this case, the master equation for $\mathbf{P}(X; t)$ takes the form

$$\frac{d}{dt}\mathbf{P}(X; t) = \mathbf{P}(x-1, x+1, x+2; t) + \mathbf{M}\,\mathbf{P}(x-1, x, x+2; t) + \mathbf{M}'\,\mathbf{P}(x-1, x, x+1; t)$$
$$+ \mathbf{M}''\,\mathbf{P}(x, x+1, x+2; t) - 3\mathbf{P}(X; t), \tag{10}$$

for some $27 \times 27$ matrices $\mathbf{M}, \mathbf{M}', \mathbf{M}''$. By the same reasoning as in case (ii), we have $\mathbf{M} = \mathbf{B} \otimes \mathbf{I}$. To determine $\mathbf{M}' = (m'_{\pi,\nu})$, note that $m'_{\pi,\nu}$ is the rate of the transition

$$(x - 1, x, x+1, \nu_1 \nu_2 \nu_3) \longrightarrow (x, x+1, x+2, \pi_1 \pi_2 \pi_3),$$

caused by the forward jump of particle $\nu_1$ over the stronger particles $\nu_2$ and $\nu_3$. This occurs through two consecutive forward jumps: first $\nu_1$ over $\nu_2$, and then over $\nu_3$, as illustrated in Figure 9. Therefore,

$$\mathbf{M}' = (\mathbf{I} \otimes \mathbf{B})(\mathbf{B} \otimes \mathbf{I}). \tag{11}$$

Figure 9: This transition occurs when $\nu_1 \leq \nu_2, \nu_3$.

To determine $\mathbf{M}'' = (m''_{\pi,\nu})$, note that $m''_{\pi,\nu}$ is the rate of the transition

$$(x, x+1, x+2, \nu_1 \nu_2 \nu_3) \longrightarrow (x, x+1, x+2, \pi_1 \pi_2 \pi_3),$$

which corresponds to an interchange of particle species without altering the position configuration. This can occur in three distinct ways:

- Interchange of the first two particles, leaving the third unchanged. This occurs when $\nu_1 > \nu_2$ and the particle of species $\nu_1$ attempts to jump. It is represented by $\mathbf{B}' \otimes \mathbf{I}$.
- Interchange of the second and third particles, leaving the first unchanged. This occurs when $\nu_2 > \nu_3$ and the particle of species $\nu_2$ attempts to jump. It is represented by $\mathbf{I} \otimes \mathbf{B}'$.
- Interchange of the first and third particles, leaving the second unchanged. This occurs when $\nu_3 < \nu_1 \leq \nu_2$ and the particle of species $\nu_1$ attempts to jump. The interchange proceeds through three consecutive moves (see Example 2.1):
  (1) $\nu_1$ jumps forward over $\nu_2$;
  (2) $\nu_1$ pushes $\nu_3$ backward;
  (3) $\nu_3$ jumps forward over $\nu_2$ to the left.
  This sequence is represented by

$$\underbrace{(\mathbf{B}' \otimes \mathbf{I})}_{(3)} \underbrace{(\mathbf{I} \otimes \mathbf{B}')}_{(2)} \underbrace{(\mathbf{B} \otimes \mathbf{I})}_{(1)}. \tag{12}$$

Combining these contributions, we obtain

$$\mathbf{M}'' = (\mathbf{B}' \otimes \mathbf{I}) + (\mathbf{I} \otimes \mathbf{B}') + (\mathbf{B}' \otimes \mathbf{I})(\mathbf{I} \otimes \mathbf{B}')(\mathbf{B} \otimes \mathbf{I}). \tag{13}$$

The last term in (13) can be expressed in an alternative form, as stated in the following lemma.

**Lemma 3.1.** Let $\mathbf{I}$ denote the $N \times N$ identity matrix, and let $\mathbf{B}, \mathbf{B}'$ be the $N^2 \times N^2$ matrices from Definition 2.2. Then, for each $N \geq 2$,

$$(\mathbf{B}' \otimes \mathbf{I})(\mathbf{I} \otimes \mathbf{B}')(\mathbf{B} \otimes \mathbf{I}) = (\mathbf{I} \otimes \mathbf{B})(\mathbf{B}' \otimes \mathbf{I})(\mathbf{I} \otimes \mathbf{B}'). \tag{14}$$

While the identity (14) can be verified directly by matrix computation, its validity for general $N$ follows naturally from the braid relation. The product

$$(\mathbf{B}' \otimes \mathbf{I})(\mathbf{I} \otimes \mathbf{B}')(\mathbf{B} \otimes \mathbf{I}) \tag{15}$$

represents three consecutive interchanges of neighboring species labels in $\nu_1 \nu_2 \nu_3$ with $\nu_3 < \nu_1 \leq \nu_2$, namely,

$$\nu_1 \nu_2 \nu_3 \;\longrightarrow\; \nu_2 \nu_1 \nu_3 \;\longrightarrow\; \nu_2 \nu_3 \nu_1 \;\longrightarrow\; \nu_3 \nu_2 \nu_1.$$

By the braid relation, the same final configuration can also be obtained via

$$\nu_1 \nu_2 \nu_3 \;\longrightarrow\; \nu_1 \nu_3 \nu_2 \;\longrightarrow\; \nu_3 \nu_1 \nu_2 \;\longrightarrow\; \nu_3 \nu_2 \nu_1,$$

which corresponds precisely to the right-hand side of (14).

**Lemma 3.2.** Let $\mathbf{I}$ denote the $N \times N$ identity matrix, and let $\mathbf{B}, \mathbf{B}'$ be the $N^2 \times N^2$ matrices from Definition 2.2. Then, for each $N \geq 2$,

$$\left( \mathbf{I}^{\otimes 3} - (\mathbf{I} \otimes \mathbf{B})(\mathbf{B}' \otimes \mathbf{I}) \right)^{-1} = \mathbf{I}^{\otimes 3} + (\mathbf{I} \otimes \mathbf{B})(\mathbf{B}' \otimes \mathbf{I}) \tag{16}$$

where

$$\mathbf{I}^{\otimes k} := \underbrace{\mathbf{I} \otimes \cdots \otimes \mathbf{I}}_{k \text{ times}}.$$

*Proof.* It suffices to show that

$$(\mathbf{I} \otimes \mathbf{B})(\mathbf{B}' \otimes \mathbf{I})(\mathbf{I} \otimes \mathbf{B})(\mathbf{B}' \otimes \mathbf{I}) = \mathbf{0}. \tag{17}$$

View $(\mathbf{I} \otimes \mathbf{B})$ and $(\mathbf{B}' \otimes \mathbf{I})$ as operators on the $N^3$-dimensional real vector space with standard basis

$$\{|\nu_1\nu_2\nu_3\rangle : \nu_i \in \{1, \ldots, N\}\},$$

ordered lexicographically from 111 to $NNN$. We must show that for all bra vectors $\langle\pi_1\pi_2\pi_3|$ and ket vectors $|\nu_1\nu_2\nu_3\rangle$,

$$\langle\pi_1\pi_2\pi_3|\,(\mathbf{I} \otimes \mathbf{B})(\mathbf{B}' \otimes \mathbf{I})(\mathbf{I} \otimes \mathbf{B})(\mathbf{B}' \otimes \mathbf{I})\,|\nu_1\nu_2\nu_3\rangle = 0. \tag{18}$$

From Definition 2.2,

$$
\begin{aligned}
(\mathbf{B}' \otimes \mathbf{I})\,|\nu_1\nu_2\nu_3\rangle &= \begin{cases} |\nu_2\nu_1\nu_3\rangle\,, & \text{if } \nu_1 > \nu_2, \\ |0\rangle\,, & \text{if } \nu_1 \leq \nu_2, \end{cases} \\[2mm]
(\mathbf{I} \otimes \mathbf{B})\,|\nu_1\nu_2\nu_3\rangle &= \begin{cases} |\nu_1\nu_3\nu_2\rangle\,, & \text{if } \nu_2 \leq \nu_3, \\ |0\rangle\,, & \text{if } \nu_2 > \nu_3, \end{cases}
\end{aligned}
\tag{19}
$$

where $|0\rangle$ denotes the zero vector. We now check that

$$(\mathbf{B}' \otimes \mathbf{I})(\mathbf{I} \otimes \mathbf{B})(\mathbf{B}' \otimes \mathbf{I})\,|\nu_1\nu_2\nu_3\rangle = |0\rangle$$

for all $|\nu_1\nu_2\nu_3\rangle$:

- If $\nu_1 \leq \nu_2$, then the first application of $(\mathbf{B}' \otimes \mathbf{I})$ yields $|0\rangle$.

- If $\nu_2 < \nu_1$ and $\nu_3 < \nu_1$, then

$$(\mathbf{I} \otimes \mathbf{B})(\mathbf{B}' \otimes \mathbf{I})\,|\nu_1\nu_2\nu_3\rangle = (\mathbf{I} \otimes \mathbf{B})\,|\nu_2\nu_1\nu_3\rangle = |0\rangle\,.$$

- If $\nu_2 < \nu_1 \leq \nu_3$, then

$$(\mathbf{B}' \otimes \mathbf{I})(\mathbf{I} \otimes \mathbf{B})(\mathbf{B}' \otimes \mathbf{I})\,|\nu_1\nu_2\nu_3\rangle = (\mathbf{B}' \otimes \mathbf{I})(\mathbf{I} \otimes \mathbf{B})\,|\nu_2\nu_1\nu_3\rangle = (\mathbf{B}' \otimes \mathbf{I})\,|\nu_2\nu_3\nu_1\rangle = |0\rangle\,.$$

This completes the proof. $\qquad\square$

**Corollary 3.3.** Let $\mathbf{I}$ denote the $N \times N$ identity matrix, and let $\mathbf{B}, \mathbf{B}'$ be the $N^2 \times N^2$ matrices from Definition 2.2. Then, for each $N \geq 2$,

$$
\begin{aligned}
\mathbf{0} &= (\mathbf{I} \otimes \mathbf{B})(\mathbf{B}' \otimes \mathbf{I})(\mathbf{I} \otimes \mathbf{B}) = (\mathbf{I} \otimes \mathbf{B}')(\mathbf{B} \otimes \mathbf{I})(\mathbf{I} \otimes \mathbf{B}') \\
&= (\mathbf{B}' \otimes \mathbf{I})(\mathbf{I} \otimes \mathbf{B})(\mathbf{B}' \otimes \mathbf{I}) = (\mathbf{B} \otimes \mathbf{I})(\mathbf{I} \otimes \mathbf{B}')(\mathbf{B} \otimes \mathbf{I}).
\end{aligned}
$$

*Proof.* The result follows by an argument analogous to that used in the proof of (17). $\qquad\square$

Let $\mathbf{U}(x_1, x_2, x_3; t) = (u_{\pi,\nu})$ denote a $27 \times 27$ matrix, where each entry $u_{\pi,\nu}$ is given by the function $U_\nu(x_1, x_2, x_3, \pi; t)$, defined for all $(x_1, x_2, x_3) \in \mathbb{Z}^3$ and $t \geq 0$.

**Proposition 3.4.** Suppose that $\mathbf{U}(x_1, x_2, x_3; t)$ satisfies

$$
\begin{aligned}
\frac{d}{dt}\mathbf{U}(x_1, x_2, x_3; t) = \; &\mathbf{U}(x_1 - 1, x_2, x_3; t) + \mathbf{U}(x_1, x_2 - 1, x_3; t) + \mathbf{U}(x_1, x_2, x_3 - 1; t) \\
&- 3\mathbf{U}(x_1, x_2, x_3; t),
\end{aligned}
\tag{20}
$$

for all $(x_1, x_2, x_3) \in \mathbb{Z}^3$, together with the boundary conditions

$$
\mathbf{U}(x, x, x'; t) = (\mathbf{B} \otimes \mathbf{I})\,\mathbf{U}(x - 1, x, x'; t) + (\mathbf{B}' \otimes \mathbf{I})\,\mathbf{U}(x, x + 1, x'; t), \tag{21}
$$

$$
\mathbf{U}(x', x, x; t) = (\mathbf{I} \otimes \mathbf{B})\,\mathbf{U}(x', x - 1, x; t) + (\mathbf{I} \otimes \mathbf{B}')\,\mathbf{U}(x', x, x + 1; t), \tag{22}
$$

for all $x, x' \in \mathbb{Z}$. Then $\mathbf{U}(x_1, x_2, x_3; t)$ satisfies all of the master equations (7), (8), (9), and (10), for their respective position configurations $(x_1, x_2, x_3)$.

*Proof.* It is immediate that $\mathbf{U}(x_1, x_2, x_3; t)$ satisfies (7) for the corresponding configurations. For the cases corresponding to (8) and (9), the result follows directly from (20) together with the boundary conditions (21) and (22).

It remains to treat the configuration $(x, x + 1, x + 2)$. In this case, (20) becomes

$$
\begin{aligned}
\frac{d}{dt}\mathbf{U}(x, x + 1, x + 2; t) = \; &\mathbf{U}(x - 1, x + 1, x + 2; t) + \mathbf{U}(x, x, x + 2; t) \\
&+ \mathbf{U}(x, x + 1, x + 1; t) - 3\mathbf{U}(x, x + 1, x + 2; t).
\end{aligned}
\tag{23}
$$

From (21) we obtain

$$
\mathbf{U}(x, x, x + 2; t) = (\mathbf{B} \otimes \mathbf{I})\,\mathbf{U}(x - 1, x, x + 2; t) + (\mathbf{B}' \otimes \mathbf{I})\,\mathbf{U}(x, x + 1, x + 2; t). \tag{24}
$$

Next, applying (22) to $\mathbf{U}(x, x + 1, x + 1; t)$ yields

$$
\mathbf{U}(x, x + 1, x + 1; t) = (\mathbf{I} \otimes \mathbf{B})\,\mathbf{U}(x, x, x + 1; t) + (\mathbf{I} \otimes \mathbf{B}')\,\mathbf{U}(x, x + 1, x + 2; t). \tag{25}
$$

Substituting (21) into the first term on the right-hand side of (25), we get

$$
\begin{aligned}
\mathbf{U}(x, x + 1, x + 1; t) = \; &(\mathbf{I} \otimes \mathbf{B})\Big[(\mathbf{B} \otimes \mathbf{I})\,\mathbf{U}(x - 1, x, x + 1; t) + (\mathbf{B}' \otimes \mathbf{I})\,\mathbf{U}(x, x + 1, x + 1; t)\Big] \\
&+ (\mathbf{I} \otimes \mathbf{B}')\,\mathbf{U}(x, x + 1, x + 2; t).
\end{aligned}
\tag{26}
$$

Rearranging terms and using Lemmas 3.1, 3.2, and Corollary 3.3, we obtain

$$
\begin{aligned}
\mathbf{U}(x, x + 1, x + 1; t) = \; &(\mathbf{I} \otimes \mathbf{B})(\mathbf{B} \otimes \mathbf{I})\,\mathbf{U}(x - 1, x, x + 1; t) \\
&+ (\mathbf{I} \otimes \mathbf{B}')\,\mathbf{U}(x, x + 1, x + 2; t) \\
&+ (\mathbf{B}' \otimes \mathbf{I})(\mathbf{I} \otimes \mathbf{B}')(\mathbf{B} \otimes \mathbf{I})\,\mathbf{U}(x, x + 1, x + 2; t).
\end{aligned}
\tag{27}
$$

Finally, substituting (24) and (27) into (23) verifies that $\mathbf{U}(x, x + 1, x + 2; t)$ satisfies (10). This completes the proof. $\qquad\square$

The outcome of this section can be summarized as follows: for the three-particle system, all master equations reduce to the evolution equation for free motion plus boundary conditions depending only on the two-particle matrices $\mathbf{B}$ and $\mathbf{B}'$. This establishes two-particle reducibility at the three-particle level.

### 3.1.2 Extension to General $n$

We now generalize to arbitrary $n \geq 2$. For an $n$-particle system, let

$$\mathbf{U}(x_1, \ldots, x_n; t) = (u_{\pi,\nu})$$

denote the $N^n \times N^n$ matrix, where each entry $u_{\pi,\nu}$ is the function $U_\nu(x_1, \ldots, x_n, \pi; t)$ defined for all $(x_1, \ldots, x_n) \in \mathbb{Z}^n$ and $t \geq 0$.

For the $N^2 \times N^2$ matrices $\mathbf{B}$ and $\mathbf{B}'$ from Definition 2.2, define for $i = 1, \ldots, n-1$,

$$\begin{aligned}
\mathbf{B}_i &:= \mathbf{I}^{\otimes(i-1)} \otimes \mathbf{B} \otimes \mathbf{I}^{\otimes(n-i-1)}, \\
\mathbf{B}'_i &:= \mathbf{I}^{\otimes(i-1)} \otimes \mathbf{B}' \otimes \mathbf{I}^{\otimes(n-i-1)},
\end{aligned} \tag{28}$$

where $\mathbf{I}$ denotes the $N \times N$ identity matrix, and define $\mathbf{B}_0 = \mathbf{B}'_0 := \mathbf{I}^{\otimes n}$ which is denoted by $\mathbb{I}$.

Proposition 3.5 below shows that for arbitrary $n$, the master equation of the system reduces to the evolution equation for non-interacting particles together with boundary conditions depending only on pairwise interactions. Hence the model satisfies two-particle reducibility.

**Proposition 3.5.** Suppose that $\mathbf{U}(x_1, \ldots, x_n; t)$ satisfies

$$\begin{aligned}
\frac{d}{dt}\mathbf{U}(x_1, \ldots, x_n; t) &= \mathbf{U}(x_1 - 1, x_2, \ldots, x_n; t) + \mathbf{U}(x_1, x_2 - 1, x_3, \ldots, x_n; t) \\
&\quad + \cdots + \mathbf{U}(x_1, x_2, \ldots, x_{n-1}, x_n - 1; t) - n\,\mathbf{U}(x_1, \ldots, x_n; t)
\end{aligned} \tag{29}$$

for all $(x_1, \ldots, x_n) \in \mathbb{Z}^n$, together with the boundary conditions

$$\begin{aligned}
\mathbf{U}(x_1, \ldots, x_{i-1}, x_i, x_i, x_{i+2}, \ldots, x_n; t) &= \mathbf{B}_i\,\mathbf{U}(x_1, \ldots, x_{i-1}, x_i - 1, x_i, x_{i+2}, \ldots, x_n; t) \\
&\quad + \mathbf{B}'_i\,\mathbf{U}(x_1, \ldots, x_{i-1}, x_i, x_i + 1, x_{i+2}, \ldots, x_n; t),
\end{aligned} \tag{30}$$

for $i = 1, \ldots, n-1$. Then $\mathbf{U}(x_1, \ldots, x_n; t)$ satisfies the master equations corresponding to each configuration $(x_1, \ldots, x_n)$ with $x_1 < \cdots < x_n$.

To prove Proposition 3.5, it suffices to verify that $\mathbf{U}(x_1, \ldots, x_n; t)$ satisfies the master equation for position configurations of the form

$$(x_1, \ldots, x_n) = (\underbrace{x_1, \ldots, x_{i-1}}_{\text{non-neighboring}}, \underbrace{x_i, x_i + 1, \ldots, x_i + l - 1}_{\text{neighboring}}, \underbrace{x_{i+l}, \ldots, x_n}_{\text{non-neighboring}})$$

for each $i = 1, \ldots, n-1$ and $l = 2, \ldots, n-i+1$, subject to

$$x_k < x_{k+1} - 1 \quad \text{for } k = 1, \ldots, i-1 \quad \text{and} \quad k = i+l-1, \ldots, n-1.$$

Equivalently, it is enough to treat the fully consecutive case

$$(x_1, \ldots, x_n) = (x, x+1, \ldots, x+n-1),$$

for each $n = 2, 3, \ldots$.

Before proving Proposition 3.5, we collect several preliminary results that will be used in the argument. These include generalizations of the lemmas established earlier for the case $n = 3$.

## Matrices for long-range interactions

Consider the transition

$$(x, x+1, \ldots, x+n-1, \nu_1 \cdots \nu_n) \longrightarrow (x+1, \ldots, x+n, \pi_1 \cdots \pi_n), \tag{31}$$

which occurs only when $\nu_1 \leq \nu_i$ for all $i = 2, \ldots, n$, and $\pi_1 \cdots \pi_n = \nu_2 \cdots \nu_n \nu_1$. This jump of the particle $\nu_1$ from $x$ to $x+n$ can be viewed as a sequence of forward jumps of $\nu_1$ over each stronger particle ahead. Consequently, the rate of the transition (31) is

$$\langle \pi_1 \cdots \pi_n | \, \mathbf{B}_{n-1} \cdots \mathbf{B}_2 \mathbf{B}_1 \, | \nu_1 \cdots \nu_n \rangle,$$

which generalizes (11).

Next, consider the transition

$$(x, x+1, \ldots, x+n-1, \nu_1 \cdots \nu_n) \longrightarrow (x, x+1, \ldots, x+n-1, \pi_1 \cdots \pi_n), \tag{32}$$

which occurs when two particles exchange their positions. In particular, let us consider the case where the first and last particles exchange positions. This swap occurs when $\nu_n < \nu_1 \leq \nu_i$ for all $i = 2, \ldots, n-1$ (see Example 2.1 for a special case). Such a long-range swap can be realized in the following order:

1. $\nu_1$ jumps forward over each stronger particle $\nu_2, \ldots, \nu_{n-1}$ sequentially;

2. $\nu_1$ pushes $\nu_n$ backward;

3. $\nu_n$ jumps forward (to the left) over each stronger particle $\nu_{n-1}, \ldots, \nu_2$ sequentially.

The corresponding transition rate is

$$\langle \pi_1 \cdots \pi_n | \, \mathbf{B}'_1 \cdots \mathbf{B}'_{n-2} \mathbf{B}'_{n-1} \, \mathbf{B}_{n-2} \cdots \mathbf{B}_1 \, | \nu_1 \cdots \nu_n \rangle, \tag{33}$$

and

$$\mathbf{B}'_1 \cdots \mathbf{B}'_{n-2} \mathbf{B}'_{n-1} \, \mathbf{B}_{n-2} \cdots \mathbf{B}_1$$

generalizes (12).

The swap of $\nu_1$ and $\nu_n$ can also be realized in an alternative order:

1. $\nu_n$ jumps forward (to the left) over the stronger particles $\nu_{n-1}, \ldots, \nu_1$ sequentially;

2. $\nu_1$ then jumps forward over the stronger particles $\nu_2, \ldots, \nu_{n-1}$ sequentially.

The corresponding transition rate is

$$\langle \pi_1 \cdots \pi_n | \, \mathbf{B}_{n-1} \cdots \mathbf{B}_2 \mathbf{B}_1' \mathbf{B}_2' \cdots \mathbf{B}_{n-1}' \, | \nu_1 \cdots \nu_n \rangle, \tag{34}$$

and

$$\mathbf{B}_{n-1} \cdots \mathbf{B}_2 \mathbf{B}_1' \mathbf{B}_2' \cdots \mathbf{B}_{n-1}' \tag{35}$$

generalizes the right-hand side of (14). Hence, we obtain the identity

$$\mathbf{B}_1' \cdots \mathbf{B}_{n-2}' \mathbf{B}_{n-1}' \, \mathbf{B}_{n-2} \cdots \mathbf{B}_1 = \mathbf{B}_{n-1} \cdots \mathbf{B}_2 \mathbf{B}_1' \mathbf{B}_2' \cdots \mathbf{B}_{n-1}', \tag{36}$$

which generalizes the identity (14).

**Master equation for** $X = (x, x+1, \ldots, x+n-1)$

Let $X_i^-$ denote the configuration obtained from

$$X = (x, x+1, \ldots, x+n-1)$$

by shifting the first $i$ coordinates one step to the left, that is,

$$X_i^- = (x-1, \, x, \, x+1, \, \ldots, \, x+i-2, \, x+i, \, x+i+1, \ldots, \, x+n-1).$$

Then the master equation for $\mathbf{P}(X;t)$ takes the form

$$\frac{d}{dt} \mathbf{P}(X;t) = \mathbf{P}(X_1^-;t) + \mathbf{M}_2 \, \mathbf{P}(X_2^-;t) + \cdots + \mathbf{M}_n \, \mathbf{P}(X_n^-;t) + \mathbf{M}_0 \, \mathbf{P}(X;t) - n \, \mathbf{P}(X;t). \tag{37}$$

Using the same reasoning as in the derivation of (11), we obtain

$$\mathbf{M}_i = \mathbf{B}_{i-1} \cdots \mathbf{B}_2 \mathbf{B}_1, \qquad i = 2, \ldots, n.$$

The entries of the matrix $\mathbf{M}_0$ describe transitions of the form

$$(X, \nu_1 \cdots \nu_n) \; \longrightarrow \; (X, \pi_1 \cdots \pi_n), \tag{38}$$

that is, swaps of two particles within the same configuration $X$.

More precisely, let $\mathbf{M}_{ij}$ with $i < j$ denote the matrix describing the interchange of the $i$th and $j$th leftmost particles. As in (35), one has

$$\mathbf{M}_{ij} = \mathbf{B}_{j-1} \cdots \mathbf{B}_{i+1} \, \mathbf{B}_i' \, \mathbf{B}_{i+1}' \cdots \mathbf{B}_{j-1}', \tag{39}$$

from which it follows that

$$\mathbf{B}_j \, \mathbf{M}_{ij} \, \mathbf{B}_j' = \mathbf{M}_{i(j+1)}. \tag{40}$$

This identity will be used later. There are $n(n-1)/2$ possible swaps of two particles, and the matrix $\mathbf{M}_0$ can be expressed as the sum of $\mathbf{M}_{ij}$,

$$\mathbf{M}_0 = \sum_{i=1}^{n-1} \sum_{j=i+1}^{n} \mathbf{M}_{ij}.$$

**Lemmas**

We now provide some technical results used to prove Proposition 3.5 for $X = (x, x+1, \ldots, x+n-1)$. The following result is a direct generalization of (17).

**Lemma 3.6.** Let $\mathbf{B}_i$ and $\mathbf{B}_i'$ be the $N^n \times N^n$ matrices defined in (28). Then, for each $k = 1, 2, \ldots, n-2$,

$$\left(\mathbf{B}_{k+1} \cdots \mathbf{B}_2 \mathbf{B}_1' \cdots \mathbf{B}_k'\right)^2 = \mathbf{0}.$$

*Proof.* We will show that

$$(\mathbf{B}_1' \cdots \mathbf{B}_k')(\mathbf{B}_{k+1} \cdots \mathbf{B}_2)(\mathbf{B}_1' \cdots \mathbf{B}_k') \, |\nu_1 \cdots \nu_n\rangle = |0\rangle \tag{41}$$

for every basis vector $|\nu_1 \cdots \nu_n\rangle$.

From the definition of $\mathbf{B}_i'$, we have

$$(\mathbf{B}_1' \cdots \mathbf{B}_k') \, |\nu_1 \cdots \nu_n\rangle = \begin{cases} |\nu_{k+1}\nu_1 \cdots \nu_k \nu_{k+2} \cdots \nu_n\rangle, & \text{if } \nu_{k+1} < \nu_i \text{ for all } i = 1, \ldots, k, \\ |0\rangle, & \text{otherwise.} \end{cases} \tag{42}$$

From the definition of $\mathbf{B}_i$, applying $(\mathbf{B}_{k+1} \cdots \mathbf{B}_2)$ to the resulting vector in (42) yields

$$(\mathbf{B}_{k+1} \cdots \mathbf{B}_2) \, |\nu_{k+1}\nu_1 \cdots \nu_k \nu_{k+2} \cdots \nu_n\rangle$$

$$= \begin{cases} |\nu_{k+1}\nu_2 \cdots \nu_k \nu_{k+2}\nu_1\nu_{k+3} \cdots \nu_n\rangle, & \text{if } \nu_1 \leq \nu_i \text{ for all } i = 2, \ldots, k+2, \\ |0\rangle, & \text{otherwise.} \end{cases} \tag{43}$$

Finally, applying $(\mathbf{B}_1' \cdots \mathbf{B}_k')$ again to the resulting vector in (43), and recalling the requirement for non-vanishing action in (42), we see that the inequality

$$\nu_{k+1} < \nu_1 \leq \nu_{k+2}$$

implied by (42) and (43) contradicts the condition for

$$(\mathbf{B}_1' \cdots \mathbf{B}_k') \, |\nu_{k+1}\nu_2 \cdots \nu_k \nu_{k+2}\nu_1\nu_{k+3} \cdots \nu_n\rangle \tag{44}$$

to be nonzero, namely $\nu_{k+2} < \nu_2, \ldots, \nu_{k+1}$. Hence (44) must equal $|0\rangle$, completing the proof. $\square$

Corollary 3.7 below generalizes Corollary 3.3, and its proof follows the same reasoning as in Lemma 3.6.

**Corollary 3.7.** We have

$$\mathbf{0} = (\mathbf{B}_{k+1} \cdots \mathbf{B}_l)(\mathbf{B}'_{l-1} \cdots \mathbf{B}'_k)(\mathbf{B}_{k+1} \cdots \mathbf{B}_l)$$

$$= (\mathbf{B}'_{l-1} \cdots \mathbf{B}'_k)(\mathbf{B}_{k+1} \cdots \mathbf{B}_l)(\mathbf{B}'_{l-1} \cdots \mathbf{B}'_k)$$

$$= (\mathbf{B}'_{k+1} \cdots \mathbf{B}'_l)(\mathbf{B}_{l-1} \cdots \mathbf{B}_k)(\mathbf{B}'_{k+1} \cdots \mathbf{B}'_l)$$

$$= (\mathbf{B}_{l-1} \cdots \mathbf{B}_k)(\mathbf{B}'_{k+1} \cdots \mathbf{B}'_l)(\mathbf{B}_{l-1} \cdots \mathbf{B}_k),$$

for each $k, l$ with $l \leq k+1$.

**Lemma 3.8.** Define matrices $\mathbf{A}_0, \mathbf{A}_1, \ldots, \mathbf{A}_{n-2}$ by

$$\mathbf{A}_0 = \mathbb{I}, \qquad \mathbf{A}_k = \mathbf{B}_{k+1}(\mathbb{I} + \mathbf{A}_{k-1})\mathbf{B}'_k \quad (k \geq 1). \tag{45}$$

Then $(\mathbf{A}_k)^2 = \mathbf{0}$ for each $k = 1, 2, \ldots, n-2$.

*Proof.* Expanding $(\mathbf{A}_k)^2$ gives

$$(\mathbf{A}_k)^2 = (\mathbf{B}_{k+1}\mathbf{B}'_k)^2 + (\mathbf{B}_{k+1}\mathbf{B}'_k)(\mathbf{B}_{k+1}\mathbf{A}_{k-1}\mathbf{B}'_k) + (\mathbf{B}_{k+1}\mathbf{A}_{k-1}\mathbf{B}'_k)(\mathbf{B}_{k+1}\mathbf{B}'_k) + (\mathbf{B}_{k+1}\mathbf{A}_{k-1}\mathbf{B}'_k)^2.$$

By Corollary 3.7 with $l = k+1$, we have

$$\mathbf{B}_{k+1}\mathbf{B}'_k\mathbf{B}_{k+1} = \mathbf{0} \quad \text{and} \quad \mathbf{B}'_k\mathbf{B}_{k+1}\mathbf{B}'_k = \mathbf{0},$$

which imply

$$(\mathbf{B}_{k+1}\mathbf{B}'_k)^2 = \mathbf{0}, \qquad (\mathbf{B}_{k+1}\mathbf{B}'_k)(\mathbf{B}_{k+1}\mathbf{A}_{k-1}\mathbf{B}'_k) = \mathbf{0}, \qquad (\mathbf{B}_{k+1}\mathbf{A}_{k-1}\mathbf{B}'_k)(\mathbf{B}_{k+1}\mathbf{B}'_k) = \mathbf{0}.$$

Thus it remains to show $(\mathbf{B}_{k+1}\mathbf{A}_{k-1}\mathbf{B}'_k)^2 = \mathbf{0}$.

Using the recursion (45),

$$\mathbf{B}_{k+1}\mathbf{A}_{k-1}\mathbf{B}'_k = \mathbf{B}_{k+1}(\mathbf{B}_k(\mathbb{I} + \mathbf{A}_{k-2})\mathbf{B}'_{k-1})\mathbf{B}'_k = \underbrace{\mathbf{B}_{k+1}\mathbf{B}_k\mathbf{B}'_{k-1}\mathbf{B}'_k}_{=:X} + \underbrace{\mathbf{B}_{k+1}\mathbf{B}_k\mathbf{A}_{k-2}\mathbf{B}'_{k-1}\mathbf{B}'_k}_{=:Y}. \tag{46}$$

Squaring and applying Corollary 3.7 (now with $l = k$) yields $X^2 = XY = YX = \mathbf{0}$, so

$$(\mathbf{B}_{k+1}\mathbf{A}_{k-1}\mathbf{B}'_k)^2 = Y^2 = \left(\mathbf{B}_{k+1}\mathbf{B}_k\mathbf{A}_{k-2}\mathbf{B}'_{k-1}\mathbf{B}'_k\right)^2.$$

Repeating this reduction step $k-1$ times gives

$$(\mathbf{B}_{k+1}\mathbf{A}_{k-1}\mathbf{B}'_k)^2 = \left(\mathbf{B}_{k+1}\mathbf{B}_k \cdots \mathbf{B}_2\,\mathbf{B}'_1 \cdots \mathbf{B}'_k\right)^2,$$

which is the zero matrix by Lemma 3.6. Hence $(\mathbf{A}_k)^2 = \mathbf{0}$. $\qquad\square$

**Corollary 3.9.** Define matrices $\mathfrak{A}_0, \mathfrak{A}_1, \ldots, \mathfrak{A}_{n-2}$ by

$$\mathfrak{A}_0 = \mathbb{I}, \qquad \mathfrak{A}_k = \left(\mathbb{I} - \mathbf{B}_{k+1}\mathfrak{A}_{k-1}\mathbf{B}'_k\right)^{-1} \quad (k \geq 1). \tag{47}$$

Then, for each $k = 1, 2, \ldots, n-2$,

$$\mathfrak{A}_k = \mathbb{I} + \mathbf{B}_{k+1}\mathfrak{A}_{k-1}\mathbf{B}'_k. \tag{48}$$

*Proof.* It suffices to show that, for every $k = 1, \ldots, n-2$,

$$\left(\mathbf{B}_{k+1}\mathfrak{A}_{k-1}\mathbf{B}'_k\right)^2 = \mathbf{0}.$$

Let $\mathbf{X}_k := \mathbf{B}_{k+1}\mathfrak{A}_{k-1}\mathbf{B}'_k$. We prove $\mathbf{X}_k^2 = \mathbf{0}$ by induction on $k$.

*Base case* $k = 1$. Here $\mathfrak{A}_0 = \mathbb{I}$, so $\mathbf{X}_1 = \mathbf{B}_2\mathbf{B}'_1$. By Corollary 3.7, we have $\left(\mathbf{B}_2\mathbf{B}'_1\right)^2 = \mathbf{0}$.

*Induction step.* Assume $\mathbf{X}_{k-1}^2 = \mathbf{0}$ for some $k \geq 2$. By the induction hypothesis and (48) at level $k-1$, we may write

$$\mathfrak{A}_{k-1} = \mathbb{I} + \mathbf{B}_k\mathfrak{A}_{k-2}\mathbf{B}'_{k-1}.$$

Hence

$$\mathbf{X}_k = \mathbf{B}_{k+1}\mathfrak{A}_{k-1}\mathbf{B}'_k = \underbrace{\mathbf{B}_{k+1}\mathbf{B}'_k}_{=:U} + \underbrace{\mathbf{B}_{k+1}\mathbf{B}_k\mathfrak{A}_{k-2}\mathbf{B}'_{k-1}\mathbf{B}'_k}_{=:V}.$$

Expanding $\mathbf{X}_k^2 = (U+V)^2$ gives $U^2 + UV + VU + V^2$. By Corollary 3.7, the terms $U^2$, $UV$, and $VU$ vanish. Thus

$$\mathbf{X}_k^2 = V^2 = \left(\mathbf{B}_{k+1}\mathbf{B}_k\mathfrak{A}_{k-2}\mathbf{B}'_{k-1}\mathbf{B}'_k\right)^2.$$

Repeating this reduction (as in the proof of Lemma 3.8) collapses the middle $\mathfrak{A}$ recursively until

$$\mathbf{X}_k^2 = \left(\mathbf{B}_{k+1}\mathbf{B}_k \cdots \mathbf{B}_2\,\mathbf{B}'_1 \cdots \mathbf{B}'_k\right)^2,$$

which equals $\mathbf{0}$ by Lemma 3.6. This completes the proof. $\qquad\square$

**Lemma 3.10.** With $\mathfrak{A}_k$ as in Corollary 3.9 and $\mathbf{M}_{ij}$ as in (39), one has, for each $k = 1, \ldots, n-1$,

$$\mathfrak{A}_{k-1}\mathbf{B}'_k = \mathbf{M}_{1(k+1)} + \cdots + \mathbf{M}_{k(k+1)}.$$

*Proof.* We argue by induction on $k$. For $k = 1$, since $\mathfrak{A}_0 = \mathbb{I}$ we have $\mathfrak{A}_0\mathbf{B}'_1 = \mathbf{B}'_1 = \mathbf{M}_{12}$, as desired.

Assume the statement holds for $k = l$ with $1 \leq l \leq n-2$, i.e.

$$\mathfrak{A}_{l-1}\mathbf{B}'_l = \mathbf{M}_{1(l+1)} + \cdots + \mathbf{M}_{l(l+1)}.$$

Using Corollary 3.9 and (40), and noting that $\mathbf{B}'_{l+1} = \mathbf{M}_{(l+1)(l+2)}$ (the case $i = l+1$, $j = l+2$ of (39)), we compute

$$\begin{aligned}
\mathfrak{A}_l\mathbf{B}'_{l+1} &= \left(\mathbb{I} + \mathbf{B}_{l+1}\mathfrak{A}_{l-1}\mathbf{B}'_l\right)\mathbf{B}'_{l+1} \\
&= \mathbf{B}'_{l+1} + \mathbf{B}_{l+1}\left(\mathbf{M}_{1(l+1)} + \cdots + \mathbf{M}_{l(l+1)}\right)\mathbf{B}'_{l+1} \\
&= \mathbf{M}_{(l+1)(l+2)} + \mathbf{M}_{1(l+2)} + \cdots + \mathbf{M}_{l(l+2)} \\
&= \mathbf{M}_{1(l+2)} + \cdots + \mathbf{M}_{(l+1)(l+2)},
\end{aligned}$$

which is exactly the claim for $k = l + 1$. This completes the induction. $\qquad\square$

The following lemma, which generalizes (27), plays a crucial role in the proof of Proposition 3.5.

**Lemma 3.11.** If the boundary condition (30) holds, then for each $l = 1, \ldots, n - 1$,

$$
\begin{aligned}
&\mathbf{U}(x, x+1, \ldots, x+l-1,\, x+l-1,\, x_{l+2}, \ldots, x_n; t) \\
&= (\mathbf{B}_l \cdots \mathbf{B}_1)\, \mathbf{U}(x-1, x, \ldots, x+l-1,\, x_{l+2}, \ldots, x_n; t) \\
&\quad + \sum_{i=1}^{l} \mathbf{M}_{i(l+1)}\, \mathbf{U}(x, x+1, \ldots, x+l-1,\, x+l,\, x_{l+2}, \ldots, x_n; t).
\end{aligned}
\tag{49}
$$

*Proof.* We argue by induction on $l$. For $l = 1$ this is precisely the boundary condition (30) with $i = 1$.

Assume (49) holds for $l = k$. Then

$$
\begin{aligned}
&\mathbf{U}(x, x+1, \ldots, x+k-1,\, x+k-1,\, x_{k+2}, \ldots, x_n; t) \\
&= (\mathbf{B}_k \cdots \mathbf{B}_1)\, \mathbf{U}(x-1, x, \ldots, x+k-1,\, x_{k+2}, \ldots, x_n; t) \\
&\quad + \sum_{i=1}^{k} \mathbf{M}_{i(k+1)}\, \mathbf{U}(x, x+1, \ldots, x+k-1,\, x+k,\, x_{k+2}, \ldots, x_n; t).
\end{aligned}
\tag{50}
$$

Apply (30) with $i = k + 1$ to

$$
\mathbf{U}(x, x+1, \ldots, x+k-1,\, x+k,\, x+k,\, x_{k+3}, \ldots, x_n; t),
$$

to get

$$
\begin{aligned}
&\mathbf{U}(x, x+1, \ldots, x+k-1,\, x+k,\, x+k,\, x_{k+3}, \ldots, x_n; t) \\
&= \mathbf{B}_{k+1}\, \mathbf{U}(x, x+1, \ldots, x+k-1,\, x+k-1,\, x+k,\, x_{k+3}, \ldots, x_n; t) \\
&\quad + \mathbf{B}'_{k+1}\, \mathbf{U}(x, x+1, \ldots, x+k-1,\, x+k,\, x+k+1,\, x_{k+3}, \ldots, x_n; t).
\end{aligned}
\tag{51}
$$

Substituting (50) into (51) and collecting terms yields

$$
\begin{aligned}
&\left[\mathbb{I} - \mathbf{B}_{k+1}\left(\sum_{i=1}^{k} \mathbf{M}_{i(k+1)}\right)\right] \mathbf{U}(x, \ldots, x+k,\, x+k,\, x_{k+3}, \ldots, x_n; t) \\
&= (\mathbf{B}_{k+1} \cdots \mathbf{B}_1)\, \mathbf{U}(x-1, x, \ldots, x+k,\, x_{k+3}, \ldots, x_n; t) \\
&\quad + \mathbf{B}'_{k+1}\, \mathbf{U}(x, \ldots, x+k,\, x+k+1,\, x_{k+3}, \ldots, x_n; t),
\end{aligned}
\tag{52}
$$

with $\mathbb{I}$ the $N^n \times N^n$ identity. By Lemma 3.10 and Corollary 3.9,

$$
\left[\mathbb{I} - \mathbf{B}_{k+1}\left(\sum_{i=1}^{k} \mathbf{M}_{i(k+1)}\right)\right]^{-1} = \mathbb{I} + \mathbf{B}_{k+1}\left(\sum_{i=1}^{k} \mathbf{M}_{i(k+1)}\right) = \mathbb{I} + \mathbf{B}_{k+1}\mathfrak{A}_{k-1}\mathbf{B}'_k.
$$

Multiplying (52) by this inverse gives

$$\mathbf{U}(x, \ldots, x+k,\, x+k,\, x_{k+3}, \ldots, x_n; t)$$
$$= \left(\mathbb{I} + \mathbf{B}_{k+1}\mathfrak{A}_{k-1}\mathbf{B}'_k\right)\left(\mathbf{B}_{k+1}\cdots\mathbf{B}_1\right)\mathbf{U}(x-1, x, \ldots, x+k,\, x_{k+3}, \ldots, x_n; t) \tag{53}$$
$$+ \left(\mathbb{I} + \mathbf{B}_{k+1}\mathfrak{A}_{k-1}\mathbf{B}'_k\right)\mathbf{B}'_{k+1}\,\mathbf{U}(x, \ldots, x+k,\, x+k+1,\, x_{k+3}, \ldots, x_n; t).$$

By repeated use of Corollaries 3.7 and 3.9,

$$\left(\mathbf{B}_{k+1}\mathfrak{A}_{k-1}\mathbf{B}'_k\right)\left(\mathbf{B}_{k+1}\cdots\mathbf{B}_1\right) = \mathbf{0}.$$

Moreover, Lemma 3.10 and (40) give

$$\left(\mathbf{B}_{k+1}\mathfrak{A}_{k-1}\mathbf{B}'_k\right)\mathbf{B}'_{k+1} = \sum_{i=1}^{k}\mathbf{M}_{i(k+2)}, \qquad \mathbf{B}'_{k+1} = \mathbf{M}_{(k+1)(k+2)}.$$

Substituting into (53) yields

$$\mathbf{U}(x, \ldots, x+k,\, x+k,\, x_{k+3}, \ldots, x_n; t)$$
$$= \left(\mathbf{B}_{k+1}\cdots\mathbf{B}_1\right)\mathbf{U}(x-1, x, \ldots, x+k,\, x_{k+3}, \ldots, x_n; t)$$
$$+ \left(\mathbf{M}_{1(k+2)} + \cdots + \mathbf{M}_{(k+1)(k+2)}\right)\mathbf{U}(x, \ldots, x+k,\, x+k+1,\, x_{k+3}, \ldots, x_n; t),$$

which is exactly (49) with $l = k+1$. The completes the proof. $\qquad\square$

**Proof of Proposition 3.5** $X = (x, x+1, \ldots, x+n-1)$

Define
$$X_{i-} := (x, x+1, \ldots, x+n-1) - (\,0, \ldots, 0, \underset{i\text{th}}{1}, 0, \ldots, 0\,),$$

and recall the notation

$$X_i^- = (x-1, x, x+1, \ldots, x+i-2, x+i, x+i+1, \ldots, x+n-1),$$

with $X_1^- = X_{1-}$.

By assumption, $\mathbf{U}(X; t)$ satisfies

$$\frac{d}{dt}\mathbf{U}(X; t) = \mathbf{U}(X_{1-}; t) + \mathbf{U}(X_{2-}; t) + \cdots + \mathbf{U}(X_{n-}; t) - n\,\mathbf{U}(X; t). \tag{54}$$

Lemma 3.11 implies that, for each $l = 2, \ldots, n$,

$$\mathbf{U}(X_{l-}; t) = \left(\mathbf{B}_{l-1}\cdots\mathbf{B}_1\right)\mathbf{U}(X_l^-; t) + \sum_{i=1}^{l-1}\mathbf{M}_{il}\,\mathbf{U}(X; t). \tag{55}$$

Substituting (55) into (54), we recover exactly equation (37). This completes the proof. $\qquad\square$

## 3.2 Yang–Baxter Integrability

The second requirement for integrability is that multi-particle scattering be consistent, i.e., that the two-body scattering matrices satisfy the Yang–Baxter relation. In earlier models such as the mTASEP and the long-range push model [13,15,16], this property follows from the specific form of the two-particle matrices. In our model, the scattering matrix takes a new algebraic form, reflecting the combined structure of $\mathbf{B}$ and $\mathbf{B}'$.

### 3.2.1 Scattering Matrix

In Section 2.3, where we analyzed the two-particle system with up to two different species, we observed that if $\mathbf{U}(x_1, x_2; t)$ satisfies (5) together with the boundary condition (6), then it also satisfies the master equation for each $(x_1, x_2)$ with $x_1 < x_2$.

Applying the Bethe ansatz yields a solution of (5) of the form, for any nonzero complex numbers $\xi_1, \xi_2$,

$$\left(\mathbf{A}_{12}\,\xi_1^{x_1}\xi_2^{x_2} + \mathbf{A}_{21}\,\xi_1^{x_2}\xi_2^{x_1}\right)e^{\varepsilon(\xi_1,\xi_2)t}, \tag{56}$$

where $\mathbf{A}_{12}$ and $\mathbf{A}_{21}$ are $4 \times 4$ constant matrices (independent of $x_1, x_2, t$), and

$$\varepsilon(\xi_1, \xi_2) = \frac{1}{\xi_1} + \frac{1}{\xi_2} - 2.$$

To enforce the boundary condition (6), substituting (56) yields

$$\mathbf{A}_{21} = \underbrace{\begin{pmatrix} -\dfrac{(1-\xi_1)\xi_2}{(1-\xi_2)\xi_1} & 0 & 0 & 0 \\ 0 & 0 & \xi_2 & 0 \\ 0 & \dfrac{1}{\xi_1} & 0 & 0 \\ 0 & 0 & 0 & -\dfrac{(1-\xi_1)\xi_2}{(1-\xi_2)\xi_1} \end{pmatrix}}_{:=\,\mathbf{R}_{21}}\mathbf{A}_{12}.$$

The matrix $\mathbf{R}_{21}$ can thus be interpreted as the two-particle scattering matrix with up to two species. We now extend this construction to the $n$-particle system with up to $N$ species.

**Definition 3.1.** Let $\mathbf{B}$ and $\mathbf{B}'$ be the $N^2 \times N^2$ matrices introduced in Definition 2.2. For any pair $(\alpha, \beta)$ with $\alpha \neq \beta$, we define the *two-particle scattering matrix (with up to $N$ species)*

$$\mathbf{R}_{\beta\alpha} := -\left(\mathbf{I} \otimes \mathbf{I} - \tfrac{\mathbf{B}}{\xi_\beta} - \mathbf{B}'\xi_\alpha\right)^{-1}\left(\mathbf{I} \otimes \mathbf{I} - \tfrac{\mathbf{B}}{\xi_\alpha} - \mathbf{B}'\xi_\beta\right), \tag{57}$$

where $\mathbf{I}$ denotes the $N \times N$ identity matrix, and we further set the *two-particle scattering matrix (with up to $N$ species)* in the $n$-particle system,

$$\mathbf{T}_{i,\beta\alpha} := \underbrace{\mathbf{I} \otimes \cdots \otimes \mathbf{I}}_{(i-1) \text{ factors}} \otimes \mathbf{R}_{\beta\alpha} \otimes \underbrace{\mathbf{I} \otimes \cdots \otimes \mathbf{I}}_{(n-i-1) \text{ factors}}$$

$$= -\left(\mathbb{I} - \frac{\mathbf{B}_i}{\xi_\beta} - \mathbf{B}'_i \xi_\alpha\right)^{-1}\left(\mathbb{I} - \frac{\mathbf{B}_i}{\xi_\alpha} - \mathbf{B}'_i \xi_\beta\right),$$

(58)

for each $i = 1, \ldots, n-1$.

### 3.2.2 Yang-Baxter Equation

Applying the Bethe ansatz, one obtains a solution of (29) of the form

$$\mathbf{U}(x_1, \ldots, x_n; t) = \sum_{\sigma \in \mathcal{S}_n} \mathbf{A}_\sigma \prod_{i=1}^{n} \xi_{\sigma(i)}^{x_i} e^{\varepsilon(\xi_i)t},$$

(59)

valid for any nonzero complex numbers $\xi_1, \ldots, \xi_n$, where the $\mathbf{A}_\sigma$ are constant $N^n \times N^n$ matrices (independent of $x_1, \ldots, x_n, t$), $\mathcal{S}_n$ is the symmetric group on $\{1, \ldots, n\}$, and

$$\varepsilon(\xi_i) = \frac{1}{\xi_i} - 1.$$

The requirement that (59) also satisfies the boundary condition (30) can be derived in the same way as for other integrable multi-species particle systems [13, 15, 16], as follows.

Let $T_i \in \mathcal{S}_n$ $(i = 1, \ldots, n-1)$ denote the adjacent transposition exchanging the $i$th and $(i+1)$th elements while leaving all others fixed. Any permutation $\sigma \in \mathcal{S}_n$ can then be expressed as a product of adjacent transpositions,

$$\sigma = T_{i_k} \cdots T_{i_1},$$

(60)

for some sequence $i_1, \ldots, i_k$ with each $i_j \in \{1, \ldots, n-1\}$.

For each $j = 1, \ldots, k$, let $(\beta_j, \alpha_j)$ denote the ordered pair of elements swapped by $T_{i_j}$, so that

$$T_{i_j}(\cdots \alpha_j \beta_j \cdots) = (\cdots \beta_j \alpha_j \cdots).$$

**Proposition 3.12.** For a permutation $\sigma = T_{i_k} \cdots T_{i_1}$, define the *multi-particle scattering matrix*

$$\mathbf{A}_\sigma := \mathbf{T}_{i_k, \beta_k \alpha_k} \cdots \mathbf{T}_{i_1, \beta_1 \alpha_1},$$

(61)

Then $\mathbf{A}_\sigma$ is well defined, that is, the product in (61) is independent of the chosen decomposition (60).

*Proof.* It suffices to show that the two-particle scattering matrices $\mathbf{T}_{i,\beta\alpha}$ satisfies

(C1) $\mathbf{T}_{i,\beta\alpha} \mathbf{T}_{j,\delta\gamma} = \mathbf{T}_{j,\delta\gamma} \mathbf{T}_{i,\beta\alpha}.$   if $|i-j| > 1$,
(C2) $\mathbf{T}_{i,\beta\alpha} \mathbf{T}_{i,\alpha\beta} = \mathbb{I}$,
(C3) $\mathbf{T}_{i+1,\gamma\beta} \mathbf{T}_{i,\gamma\alpha} \mathbf{T}_{i+1,\beta\alpha} = \mathbf{T}_{i,\beta\alpha} \mathbf{T}_{i+1,\gamma\alpha} \mathbf{T}_{i,\gamma\beta}.$

The identities (C1) and (C2) are immediate from (58). For (C3), the required relation is precisely the Yang-Baxter equation for the two-particle scattering matrices,

$$(\mathbf{R}_{\gamma\beta} \otimes \mathbf{I})(\mathbf{I} \otimes \mathbf{R}_{\gamma\alpha})(\mathbf{R}_{\beta\alpha} \otimes \mathbf{I}) = (\mathbf{I} \otimes \mathbf{R}_{\beta\alpha})(\mathbf{R}_{\gamma\alpha} \otimes \mathbf{I})(\mathbf{I} \otimes \mathbf{R}_{\gamma\beta}), \tag{62}$$

for distinct labels $\alpha, \beta, \gamma$.

Since only the three labels $\alpha, \beta, \gamma$ are involved, the general $N$ case reduces to checking (62) for $N = 3$, that is, a $27 \times 27$ matrix identity. This is the same reduction as detailed in Lemma A.2 of [15], and the verification can be done directly. $\qquad\square$

The proof of Proposition 3.13 below is essentially the same as that of Proposition 3.1 in [16], with our two-particle scattering matrix $\mathbf{R}_{\beta\alpha}$ from (57) replacing the one used there. For completeness and self-containedness, however, we provide the proof in a concise way.

**Proposition 3.13.** Let $\mathbf{A}_\sigma$ be defined as in (61). Then the Bethe ansatz solution (59) satisfies the boundary conditions (30).

*Proof.* Fix $i \in \{1, \ldots, n-1\}$ and consider the boundary condition (30). Substituting the Bethe ansatz solution (59) into (30) and collecting monomials in the spectral variables yields

$$\sum_{\sigma \in \mathcal{S}_n} \left( \mathbb{I} - \tfrac{\mathbf{B}_i}{\xi_{\sigma(i)}} - \mathbf{B}_i' \, \xi_{\sigma(i+1)} \right) \mathbf{A}_\sigma = \mathbf{0}. \tag{63}$$

Partition $\mathcal{S}_n$ into disjoint pairs $\{\sigma, T_i\sigma\}$. For each such pair, set $\alpha = \sigma(i)$ and $\beta = \sigma(i+1)$. By construction of $\mathbf{A}_\sigma$ and (58), we have

$$\mathbf{A}_{T_i\sigma} = \mathbf{T}_{i,\beta\alpha} \, \mathbf{A}_\sigma = -\left( \mathbb{I} - \tfrac{\mathbf{B}_i}{\xi_\beta} - \mathbf{B}_i' \, \xi_\alpha \right)^{-1} \left( \mathbb{I} - \tfrac{\mathbf{B}_i}{\xi_\alpha} - \mathbf{B}_i' \, \xi_\beta \right) \mathbf{A}_\sigma.$$

Consequently,

$$\left( \mathbb{I} - \tfrac{\mathbf{B}_i}{\xi_\alpha} - \mathbf{B}_i' \, \xi_\beta \right) \mathbf{A}_\sigma + \left( \mathbb{I} - \tfrac{\mathbf{B}_i}{\xi_\beta} - \mathbf{B}_i' \, \xi_\alpha \right) \mathbf{A}_{T_i\sigma} = \mathbf{0}.$$

Summing this identity over all pairs $\{\sigma, T_i\sigma\}$ establishes (63). Hence the Bethe ansatz solution satisfies the boundary condition for each $i$, as required. $\qquad\square$

## 3.3 Transition Probabilities

We now derive explicit formulas for the transition probabilities. Up to this point, we have shown that the Bethe ansatz solution (59), with $\mathbf{A}_\sigma$ defined as in (61), satisfies the master equation governing $\mathbf{P}_Y(X; t)$ for every *physical* configuration $X$. However, the expression (59) does not yet coincide with the transition probability matrix $\mathbf{P}_Y(X; t)$, since the initial condition has not been enforced. This initial condition is given by

$$\mathbf{P}_Y(X; 0) = \begin{cases} \mathbb{I}, & \text{if } X = Y, \\ \mathbf{0}, & \text{otherwise.} \end{cases} \tag{64}$$

where $\mathbf{0}$ denotes the $N^n \times N^n$ zero matrix.

As in earlier works on integrable models (see, e.g., [13, 23]), we impose the initial condition by taking contour integrals of the Bethe ansatz solution (59), multiplied by $\prod_i \xi_i^{-y_i - 1}$, over suitable contours. This standard procedure ensures that the required initial condition is satisfied and yields an explicit integral representation of the transition probabilities. The resulting formula has the same overall structure as in earlier models such as the mTASEP, though the entries of $\mathbf{A}_\sigma$ take a different algebraic form in the present case. We now state the result precisely.

**Theorem 3.14** (Transition probability of the mTASEP with long-range swap)**.** Let $\mathbf{A}_\sigma$ be defined as in (61). Then the matrix of transition probabilities $\mathbf{P}_Y(X; t)$ admits the contour-integral representation

$$\mathbf{P}_Y(X; t) = \oint_C \cdots \oint_C \sum_{\sigma \in \mathcal{S}_n} \mathbf{A}_\sigma \prod_{i=1}^n \left( \xi_{\sigma(i)}^{x_i - y_{\sigma(i)} - 1} \, e^{\varepsilon(\xi_i) t} \right) d\xi_1 \cdots d\xi_n, \tag{65}$$

where $C$ is a positively oriented circle centered at the origin of radius larger than 1, $\oint = (1/2\pi i) \int$, and the integrals are taken entrywise. Here $\varepsilon(\xi) = \xi^{-1} - 1$, as in (59). Hence, the formula of transition probability $P_{(Y,\nu)}(X, \pi; t)$ is given by

$$P_{(Y,\nu)}(X, \pi; t) = \oint_C \cdots \oint_C \sum_{\sigma \in \mathcal{S}_n} (\mathbf{A}_\sigma)_{\pi,\nu} \prod_{i=1}^n \left( \xi_{\sigma(i)}^{x_i - y_{\sigma(i)} - 1} \, e^{\varepsilon(\xi_i) t} \right) d\xi_1 \cdots d\xi_n.$$

*Proof sketch.* Since the Bethe ansatz solution—which appears as the integrand of (65)—already satisfies the master equation for each physical configuration $X$ by construction, the contour integral does so as well. Thus, it remains only to verify the initial condition. At $t = 0$, the contribution from the identity permutation is

$$\oint_C \cdots \oint_C \mathbf{A}_{\mathrm{id}} \prod_{i=1}^n \xi_i^{x_i - y_i - 1} \, d\xi_1 \cdots d\xi_n = \begin{cases} \mathbb{I}, & \text{if } X = Y, \\ \mathbf{0}, & \text{otherwise,} \end{cases}$$

since $\mathbf{A}_{\mathrm{id}} = \mathbb{I}$ for the identity permutation and by the residue theorem. Hence, it remains to show

$$\oint_C \cdots \oint_C \sum_{\sigma \neq \mathrm{id}} \mathbf{A}_\sigma \prod_{i=1}^n \xi_i^{x_i - y_i - 1} \, d\xi_1 \cdots d\xi_n = \mathbf{0}$$

for any $X = (x_1, \ldots, x_n)$ and $Y = (y_1, \ldots, y_n)$ with $x_i < x_{i+1}$, $y_i < y_{i+1}$ and $y_i \leq x_i$ for all $i$. Indeed, we can show that the integral is zero for each non-identity permutation. The detailed argument is provided in Appendix. $\qquad \square$

# 4    Discussion

Our study reveals several structural features of the long-range swap model and its connections to other integrable multispecies exclusion processes.

## Coupling perspective

Both the mTASEP and the TASEP with long-range push [15] can be realized as couplings of single-species models with different initial conditions, where discrepancies evolve as lower-ranked particles. For the long-range swap dynamics, however, no such coupling representation appears to exist. This absence of a natural single-species coupling highlights a key distinction from earlier models.

## Symmetry in boundary conditions and two integrable types

In the formulation studied in Section 2, we have seen that when two particles of the same species meet, their interaction follows the drop–push rule, leading to the boundary relation

$$\mathbf{U}(x,x;t) = \begin{pmatrix} 1 & 0 & 0 & 0 \\ 0 & 0 & 0 & 0 \\ 0 & 1 & 0 & 0 \\ 0 & 0 & 0 & 1 \end{pmatrix} \mathbf{U}(x-1,x;t) + \begin{pmatrix} 0 & 0 & 0 & 0 \\ 0 & 0 & 1 & 0 \\ 0 & 0 & 0 & 0 \\ 0 & 0 & 0 & 0 \end{pmatrix} \mathbf{U}(x,x+1;t). \tag{66}$$

If instead we assume that identical species interact as in the ordinary TASEP when they meet, the boundary condition becomes

$$\mathbf{U}(x,x;t) = \begin{pmatrix} 0 & 0 & 0 & 0 \\ 0 & 0 & 0 & 0 \\ 0 & 1 & 0 & 0 \\ 0 & 0 & 0 & 0 \end{pmatrix} \mathbf{U}(x-1,x;t) + \begin{pmatrix} 1 & 0 & 0 & 0 \\ 0 & 0 & 1 & 0 \\ 0 & 0 & 0 & 0 \\ 0 & 0 & 0 & 1 \end{pmatrix} \mathbf{U}(x,x+1;t). \tag{67}$$

The symmetry between the matrices in (66) and (67) suggests a flexible framework in which the interaction of identical species can be tuned. Importantly, integrability of the model with the boundary condition (67) can be established in essentially the same way as in the present work: both the two-body reducibility and the Yang–Baxter equation can be verified by analogous arguments. For clarity, we therefore refer to the first case as the *mTASEP with long-range swap (drop–push type)* and to the second as the *mTASEP with long-range swap (TASEP type)*. (The classical mTASEP may be referred to as the *short-range swap model*.)

## Connections with classical models

If all particles belong to the same species, the TASEP-type long-range swap reduces to the classical TASEP, while the drop–push-type long-range swap reduces to the drop–push model. Thus the TASEP has two multi-species extensions: the mTASEP with short-range swap (that is, the classical mTASEP) and the mTASEP with long-range swap (TASEP type). Similarly, the drop–push model has two multi-species extensions: the mTASEP with long-range push [15] and the mTASEP with long-range swap (drop–push type).

| TASEP | |
|---|---|
| Multi-species version I | mTASEP (short-range swap) |
| Multi-species version II | mTASEP with long-range swap (TASEP type) |

| Drop–push model | |
|---|---|
| Multi-species version I | mTASEP with long-range push |
| Multi-species version II | mTASEP with long-range swap (drop–push type) |

Table 1: Classification of multispecies extensions of TASEP and the drop–push model.

### Non-integrable alternative

The dynamics studied in this work were defined by the rule that stronger particles push weaker ones backward, while weaker particles may jump forward over stronger ones. By contrast, if one reverses this interaction rule—so that stronger particles push weaker ones forward while weaker particles are pushed backward—the resulting process fails to define a consistent dynamics due to the emergence of infinite loops.

For example, consider particles of species 2, 1, and 3 occupying sites $x$, $x+1$, and $x+2$, respectively. If the particle of species 2 jumps to $x+1$, it pushes the particle of species 1 forward to $x+2$. But once at $x+2$, the particle of species 1 encounters the stronger particle of species 3 and is forced to jump backward to $x+1$. Since $x+1$ is still occupied by the particle of species 2, this cycle repeats indefinitely, producing an endless back-and-forth motion. Such loops prevent the construction of a well-defined $n$-particle Markovian dynamics.

In this case, the corresponding boundary condition takes the form

$$
\mathbf{U}(x,x;t) = \begin{pmatrix} 0 & 0 & 0 & 0 \\ 0 & 0 & 0 & 0 \\ 0 & 0 & 1 & 0 \\ 0 & 0 & 0 & 0 \end{pmatrix} \mathbf{U}(x-1,x;t) + \begin{pmatrix} 1 & 0 & 0 & 0 \\ 0 & 1 & 0 & 0 \\ 0 & 0 & 0 & 0 \\ 0 & 0 & 0 & 1 \end{pmatrix} \mathbf{U}(x,x+1;t). \tag{68}
$$

Interestingly, the two-particle scattering matrix derived from (68) still satisfies the Yang-Baxter equation. However, for $n$-particle systems with $n \geq 3$, the key reducibility property (Proposition 3.5) fails, and thus the model cannot be regarded as integrable.

## 5 Summary and Outlook

In this work, we introduced and analyzed the multispecies totally asymmetric simple exclusion process with long-range swap, a new interacting particle system that combines the backward-push

rule of the mTASEP with the forward-jump rule of the long-range push model. Although the dynamics appear at first glance to involve only local interactions, they naturally generate effective long-range exchanges of particles. We established integrability of the model by proving two-particle reducibility and showing that the associated scattering matrix satisfies the Yang-Baxter equation. In addition, we derived explicit contour-integral formulas for the transition probabilities, thereby placing this model as a new member of the class of exactly solvable multispecies processes.

Our analysis highlighted several distinctive features. In particular, the boundary conditions display a symmetry that leads naturally to two types of long-range swap models. Unlike the mTASEP and the long-range push model, however, the dynamics introduced here do not admit a straightforward coupling interpretation in terms of single-species systems, underscoring their novelty.

This study also raises a number of open problems and potential directions for future work. From a probabilistic perspective, one may investigate asymptotic properties such as current fluctuations and scaling limits within the KPZ universality class, especially for the TASEP-type long-range swap model under special initial conditions. For example, one can ask how the behavior of species-1 particles deviates from that in the short-range swap setting (that is, the classical mTASEP), and how it compares with known results for the classical mTASEP [1,6,18,24]. From a modeling standpoint, it is natural to ask whether interpolations between the two types of long-range swap introduced here, or extensions to partially asymmetric dynamics, preserve integrability. More broadly, it remains open whether alternative choices of $\mathbf{B}$ and $\mathbf{B}'$ could yield additional integrable models. On the algebraic side, it would be valuable to clarify the relation of the scattering matrices derived here to the stochastic six-vertex model [8]. More generally, it is an open problem to systematically characterize which hybrid interaction rules lead to solvable dynamics, and to determine whether additional families of integrable models exist beyond the known extremes of backward-push and forward-push interactions.

In summary, the long-range swap model broadens the family of integrable interacting particle systems and opens new directions at the intersection of integrable probability and statistical mechanics.

**Acknowledgement.** This research was funded by Nazarbayev University under the Faculty-Development Competitive Research Grants Program for 2024–2026 (grant number 201223FD8822).

# A   Appendix: Proof of Theorem 3.14

We complete the proof of Theorem 3.14 by proving Proposition A.1 below.

**Definition A.1.** An *inversion* of a permutation $\sigma = \sigma(1)\sigma(2)\cdots\sigma(n)$ is a pair $(\sigma(i), \sigma(j))$ such that $\sigma(i) > \sigma(j)$ and $i < j$. Let $\mathrm{Inv}(\sigma)$ denote the set of all inversions in $\sigma$.

**Proposition A.1.** Suppose that $X = (x_1, \ldots, x_n)$ and $Y = (y_1, \ldots, y_n)$ satisfy $x_i < x_{i+1}$, $y_i < y_{i+1}$, and $y_i \leq x_i$ for all $i$. Then, for any non-identity permutation $\sigma$,

$$\oint_C \cdots \oint_C \mathbf{A}_\sigma \prod_{i=1}^n \xi_{\sigma(i)}^{x_i - y_{\sigma(i)} - 1} \, d\xi_1 \cdots d\xi_n = \mathbf{0},$$

where $C$ is a positively oriented circle centered at the origin with radius larger than 1.

*Proof.* Fix $\sigma \neq \mathrm{id}$. From the definition (61), each entry $(\mathbf{A}_\sigma)_{\pi,\nu}$ is a finite sum of terms of the form

$$\prod_{(\beta,\alpha)\in\mathrm{Inv}(\sigma)} R_{\beta\alpha}, \tag{69}$$

and for each inversion $(\beta, \alpha)$ one factor $R_{\beta\alpha}$ is chosen from

$$-\frac{(1-\xi_\alpha)\xi_\beta}{(1-\xi_\beta)\xi_\alpha}, \qquad \frac{1}{\xi_\alpha}, \qquad \xi_\beta, \qquad 0.$$

Any term with a zero factor vanishes. Assume henceforth that all selected $R_{\beta\alpha} \neq 0$. We will show that in every such case the integral is zero.

*Case 1: $X \neq Y$.* Pick $i_0$ with $x_{i_0} \neq y_{i_0}$. Choose $i$ so that $\sigma(i) \leq i_0 \leq i$ and $\sigma(i) = \min\{\sigma(k) : k \geq i\}$. Then all elements smaller than $\sigma(i)$ lie to the left of $\sigma(i)$. Thus there are exactly $i - \sigma(i)$ inversions of the form $(\beta, \sigma(i))$ and none of the form $(\sigma(i), \alpha)$. Consequently, no factor $(1 - \xi_{\sigma(i)})^{-1}$ appears in (69), and the exponent of $\xi_{\sigma(i)}$ in the full integrand is bounded below by

$$x_i - y_{\sigma(i)} - 1 + \sigma(i) - i \geq 0,$$

since $x_i - y_{\sigma(i)} \geq i - \sigma(i) + 1$ from $y_{\sigma(i)} \leq y_{i_0} < x_{i_0} \leq x_i$. Hence the integrand is analytic in $\xi_{\sigma(i)}$, and the $\xi_{\sigma(i)}$-integral over $C$ vanishes by Cauchy's theorem.

*Case 2: $X = Y$.* Choose $i$ such that $\sigma(i) > i$ and $\sigma(i) = \max\{\sigma(k) : k \leq i\}$. Then all elements larger than $\sigma(i)$ lie to the right of $\sigma(i)$. Thus there are $\sigma(i) - i$ inversions of the form $(\sigma(i), \alpha)$ and none of the form $(\beta, \sigma(i))$. Consequently, the factor involving $\xi_{\sigma(i)}$ in (69) multiplied by $\xi_{\sigma(i)}^{y_i - y_{\sigma(i)} - 1}$ is of the form

$$\left( -\frac{\xi_{\sigma(i)}}{1 - \xi_{\sigma(i)}} \right)^a \cdot (\xi_{\sigma(i)})^{b + y_i - y_{\sigma(i)} - 1}, \qquad a + b \leq \sigma(i) - i, \tag{70}$$

for some nonnegative integers $a, b$. Thus the $\xi_{\sigma(i)}$-dependence of the integrand is a rational function $P(\xi_{\sigma(i)})/Q(\xi_{\sigma(i)})$ with

$$\deg Q - \deg P = y_{\sigma(i)} - y_i + 1 - b \geq \sigma(i) - i + 1 - (\sigma(i) - i) \geq 1.$$

If $b < \sigma(i) - i$, then $\deg Q - \deg P \geq 2$, so the integrand is $O(1/\xi_{\sigma(i)}^2)$ as $|\xi_{\sigma(i)}| \to \infty$ and the contour integral over $C$ vanishes.

It remains to consider the borderline case $b = \sigma(i) - i$, i.e. when every factor $R_{\sigma(i)\alpha}$ contributes only $\xi_{\sigma(i)}$. In this situation, select $j > i$ such that $\sigma(j) < j$, $\sigma(i) > \sigma(j)$, and $\sigma(j) = \min\{\sigma(k) : k \geq j\}$. (This is possible since $\sigma \neq \mathrm{id}$.) Then all elements smaller than $\sigma(j)$ lie to the left of $\sigma(j)$, so there are $j - \sigma(j)$ inversions of the form $(\beta, \sigma(j))$ and none of the form $(\sigma(j), \alpha)$. Consequently, the factor involving $\xi_{\sigma(j)}$ in (69) multiplied by $\xi_{\sigma(j)}^{y_j - y_{\sigma(j)} - 1}$ is of the form

$$\left( -\frac{1 - \xi_{\sigma(j)}}{\xi_{\sigma(j)}} \right)^a \left( \frac{1}{\xi_{\sigma(j)}} \right)^b, \qquad a + b \leq j - \sigma(j) - 1, \tag{71}$$

where $-1$ in $j - \sigma(j) - 1$ is due to that for the inversion $(\sigma(i), \sigma(j))$, $R_{\sigma(i)\sigma(j)}$ takes $\xi_{\sigma(i)}$. Hence the exponent of $\xi_{\sigma(j)}$ in the full integrand is bounded below by

$$y_j - y_{\sigma(j)} - 1 - (j - \sigma(j) - 1) \ \geq \ (j - \sigma(j)) - 1 - (j - \sigma(j) - 1) \ \geq \ 0,$$

so $\xi_{\sigma(j)}$ has no pole. Therefore the contour integral in $\xi_{\sigma(j)}$ also vanishes. $\qquad\square$

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
