# Peer review of "Multispecies totally asymmetric simple exclusion process with long-range swap"

_SciPost Physics Core_

## Round 1 · Referee Report · Anonymous (Referee 1) · 2025-11-17

Strengths

1- Clean final result, eq. (65), (61), from which asymptotic analysis appears plausible.

Weaknesses

1- Yet another integrable variant of the exclusion process, without a clear physical motivation, such as whether a new large scale behaviour is expected compared to earlier models.

Report

The author introduces a new variant of the multispecies totally asymmetric simple exclusion process, an integrable model of hard-core particles moving in one dimension, with additional long range swap of particles. He obtains in particular an exact Bethe ansatz formula for the propagator of the model defined on the infinite line, involving contour integrals on the Bethe rapidities.

The paper is reasonably well written, with an introduction explaining the dynamics of the model in a clear way.

The paper is however quite technical. Additionally, the paper ultimately deals only with some variant of a model solved before, with a final result quite similar in structure to earlier results for other kinds of multispecies exclusion processes. As such, I think that the paper would be a better fit for SciPost Physics Core.

Requested changes

1- It would be useful to write explicitly how the coefficients A_sigma in your case differ from those of the earlier multispecies TASEP.

2-Multispecies exclusion processes are related to q-deformed SU(N) quantum groups, and a nested Bethe ansatz procedure is generally used to diagonalize the generator of the evolution. On the other hand, your formula (65) for the propagator involves a single set of Bethe rapidities. Can you comment on this ?

Recommendation

Accept in alternative Journal (see Report)

  • validity: top
  • significance: good
  • originality: good
  • clarity: high
  • formatting: perfect
  • grammar: perfect

Author:  Eunghyun Lee  on 2025-12-17  [id 6152]

(in reply to Report 1 on 2025-11-17)

Response to Comment 1:

We thank the referee for this suggestion.

We have clarified this point in Section~3.2 of the revised manuscript. In particular,
in Section~3.2.1, we now explicitly display the two-particle scattering matrix for the present model and compare it with the corresponding matrix for the usual multispecies TASEP; after the full construction of the multi-particle coefficients
$\mathbf{A}_\sigma$, we added a remark (Remark 3.3) emphasizing that, unlike the triangular scattering matrices of the usual multispecies TASEP reflecting priority-based interactions, the scattering matrices in our model involve both
$\mathbf{B}$ and $\mathbf{B}'$ and encode nontrivial long-range swap processes. As a consequence, the resulting coefficients $\mathbf{A}_\sigma$ differ qualitatively from those appearing in earlier multispecies exclusion processes.

Response to Comment 2:

We appreciate this insightful comment. We added a remark (Remark 3.2) immediately after the Bethe ansatz formula for the propagator (Eq.~(65) in the revised manuscript) clarifying this point. In our approach, the Bethe ansatz is applied to the
spatial evolution in order to construct the transition probabilities, rather than to a spectral diagonalization of the generator. The dependence on particle species is encoded in the matrix-valued amplitudes $\mathbf{A}_\sigma$, which
are constructed from the two-particle scattering matrices $\mathbf{R}_{\beta\alpha}$ and $\mathbf{T}_{i,\beta\alpha}$. The consistency of this construction follows from two-particle reducibility and the Yang--Baxter relations established in Section~3.2. This explains why a single set of spectral parameters is sufficient in our formulation.

---

## Round 1 · Referee Report · Anonymous (Referee 2) · 2025-11-26

Strengths

1 New integrable variant of the TASEP with an effective long-range interaction
2 Explicit contour integral formulas for transition probabilities

Weaknesses

1 No asymptotic results for large times
2 No physics motivation for introducing this model

Report

The paper introduces an interesting integrable variant of the TASEP which despite local moves realize an effective long-range interaction. This is important as - in principle - it allows to address by exact methods the generally open question whether or not long-range interactions lead to different dynamical universality classes. The paper provides a first step towards answering this question for this specific model. It would be desirable to have this issue discussed at least briefly as a motivation for introducing this model. Some indications as to why this is relevant can be found e.g. in a PRL by Karevski and Schuetz [Karevski D and Schütz G M 2017 Conformal invariance in driven diffusive systems at high currents PRL 118 030601] where local moves with long-range interactions in a different integrable TASEP give rise to different large-scale behaviour,
including a dynamical phase transition discussed in generic terms in A. Lazarescu, Generic dynamical phase transition in one-dimensional bulk-driven lattice gases with exclusion, J. Phys. A 50, 254004 (2017).
With such a revision I would recommend publication, even though I feel that the paper might be more appropriate for SciPost core.

Requested changes

1 Provide a physical motivation for introducing this model, e.g. by discussing the two papers mentioned in the report.

Recommendation

Ask for minor revision

  • validity: top
  • significance: good
  • originality: high
  • clarity: top
  • formatting: perfect
  • grammar: excellent

Author:  Eunghyun Lee  on 2025-12-17  [id 6153]

(in reply to Report 2 on 2025-11-26)

We thank the referee for the careful reading of the manuscript and for the constructive suggestion regarding the physical motivation of the model.

We have added a new motivation paragraph near the end of the Introduction. There we explain that one of the motivations for introducing the present model is to investigate whether effective long-range interactions, even when generated by purely local update rules, can lead to new large-scale dynamical behavior. As suggested by the referee, we now explicitly cite and briefly discuss related works by Karevski and Sch\"{u}tz (Phys.\ Rev.\ Lett.\ \textbf{118}, 030601 (2017)) and Lazarescu (J.\ Phys.\ A:\ Math.\ Theor.\ \textbf{50}, 254004 (2017)), and we clarify that the present work focuses on integrability and exact transition probabilities, while leaving large-time asymptotic analysis for future investigation.

---

## Round 2 · Referee Report · Anonymous (Referee 2) · 2025-12-17

Report

The revision is fine. The manuscript can be published.

Recommendation

Publish (meets expectations and criteria for this Journal)

---

## Round 2 · Referee Report · Anonymous (Referee 1) · 2025-12-18

Report

The author has satisfactorily addressed the comments of the referees. I recommend the paper for publication in Scipost.

Recommendation

Publish (easily meets expectations and criteria for this Journal; among top 50%)

---

## Round 2 · Author Response

We thank you for the careful evaluation of our manuscript, as well as for the opportunity to resubmit it to SciPost Physics Core. We are grateful to both referees for their constructive comments and positive assessments. We have revised the manuscript accordingly and believe that all concerns raised in the reports have now been fully addressed.

---

## Round 2 · List of Changes

Response to Referee 1:

Response to Comment 1: We thank the referee for this suggestion. We have clarified this point in Section~3.2 of the revised manuscript. In particular, in Section~3.2.1, we now explicitly display the two-particle scattering matrix for the present model and compare it with the corresponding matrix for the usual multispecies TASEP;
after the full construction of the multi-particle coefficients $\mathbf{A}_\sigma$, we added a remark (Remark 3.3) emphasizing that, unlike the triangular
scattering matrices of the usual multispecies TASEP reflecting priority-based interactions, the scattering matrices in our model involve both $\mathbf{B}$ and $\mathbf{B}'$ and encode nontrivial long-range swap processes. As a consequence, the resulting coefficients $\mathbf{A}_\sigma$ differ qualitatively from those appearing in earlier multispecies exclusion processes.

Response to Comment 2: We appreciate this insightful comment. We added a remark (Remark 3.2) immediately after the Bethe ansatz formula for the propagator (Eq.~(65) in the revised manuscript) clarifying this point. In our approach, the Bethe ansatz is applied to the spatial evolution in order to construct the transition probabilities, rather than to a spectral diagonalization of the generator. The dependence on particle species is encoded in the matrix-valued amplitudes $\mathbf{A}_\sigma$, which are constructed from the two-particle scattering matrices $\mathbf{R}_{\beta\alpha}$ and $\mathbf{T}_{i,\beta\alpha}$. The consistency of this construction follows from two-particle reducibility and the Yang--Baxter relations established in Section~3.2. This explains why a single set of spectral parameters is sufficient in our formulation.

Response to Referee 2:

We thank the referee for this suggestion. We have added a new motivation paragraph near the end of the Introduction. There we explain that one of the motivations for introducing the present model is to investigate whether effective long-range interactions, even when generated by purely local update rules, can lead to new large-scale dynamical behavior. As suggested by the referee, we now explicitly cite and briefly discuss related works by Karevski and Sch\"{u}tz (Phys.\ Rev.\ Lett.\ \textbf{118}, 030601 (2017)) and Lazarescu (J.\ Phys.\ A:\ Math.\ Theor.\ \textbf{50}, 254004 (2017)), and we clarify that the present work focuses on integrability and exact transition probabilities, while leaving large-time asymptotic analysis for future investigation.

---

## Editorial Decision

accepted_in_target_journal